# Adenosine Induces EBV Lytic Reactivation through ADORA1 in EBV-Associated Gastric Carcinoma

**DOI:** 10.3390/ijms20061286

**Published:** 2019-03-14

**Authors:** Su Jin Choi, Eunhyun Ryu, Seulki Lee, Sora Huh, Yu Su Shin, Byung Woog Kang, Jong Gwang Kim, Hyosun Cho, Hyojeung Kang

**Affiliations:** 1College of Pharmacy and Cancer Research Institute and Research Institute of Pharmaceutical Sciences, Kyungpook National University, Daegu 41566, Korea; sujinchoi88@naver.com (S.J.C.); fbdmsgus@naver.com (E.R.); hsl367@naver.com (S.H.); 2College of Pharmacy and Innovative Drug Center, Duksung Women’s University, Seoul 01369, Korea; mchild1987@naver.com; 3Department of Medical Crop Research, National Institute of Horticultural and Herbal Science, Rural Development Administration, Eumseong 27709, Korea; totoro69@korea.kr; 4Department of Oncology/Hematology and Cancer Research Institute and School of Medicine, Kyungpook National University Hospital and Kyungpook National University, Daegu 41404, Korea; bwkang@knu.ac.kr (B.W.K.); jkk21c@knu.ac.kr (J.G.K.)

**Keywords:** *Cordyceps militaris* extracts, Adenosine, Epstein-Barr virus, EBV-associated gastric carcinoma, ADORA1

## Abstract

*Cordyceps* species are known to contain numerous bioactive compounds, including cordycepin. Extracts of *Cordyceps militaris* (CME) are used in diverse medicinal purposes because of their bioactive components. Cordycepin, one of the active components of CME, exhibits anti-proliferative, pro-apoptotic, and anti-inflammatory effects. Cordycepin structurally differs from adenosine in that its ribose lacks an oxygen atom at the 3′ position. We previously reported that cordycepin suppresses Epstein–Barr virus (EBV) gene expression and lytic replication in EBV-associated gastric carcinoma (EBVaGC). However, other studies reported that cordycepin induces EBV gene expression and lytic reactivation. Thus, it was reasonable to clarify the bioactive effects of CME bioactive compounds on the EBV life cycle. We first confirmed that CME preferentially induces EBV gene expression and lytic reactivation; second, we determined that adenosine in CME induces EBV gene expression and lytic reactivation; third, we discovered that the adenosine A1 receptor (ADORA1) is required for adenosine to initiate signaling for upregulating *BZLF1,* which encodes for a key EBV regulator (Zta) of the EBV lytic cycle; finally, we showed that *BZLF1* upregulation by adenosine leads to delayed tumor development in the EBVaGC xenograft mouse model. Taken together, these results suggest that adenosine is an EBV lytic cycle inducer that inhibits EBVaGC development.

## 1. Introduction

Epstein–Barr virus (EBV), is a human gamma-herpesvirus that establishes lifelong infections in more than 95% of the human population [1]. EBV causes infectious mononucleosis, and EBV-related malignancies, such as Burkitt’s lymphoma (BL), nasopharyngeal carcinoma (NPC), and EBV-associated gastric carcinoma (EBVaGC) [2]. Khan et al. reported that 1.8% of all cancer deaths represented EBV associated malignancies cases in 2010 worldwide and the largest mortality from EBV associated malignancies was for EBVaGC from 1990 to 2010 (69,081 cases) [3]. EBVaGC accounts for averagely 10% in all gastric cancer cases and the incidence of EBVaGC depends on geographic distribution and environmental factors [3,4]. EBVaGC has distinct clinicopathological features which are lymphoepithelioma-like carcinoma with extremely dense lymphocyte infiltration [5]. 

Like all herpesviruses, EBV can infect cells in either a latent or lytic form. Latent infection occurs in memory B cells, allowing the virus to evade the host immune response and persist indefinitely within humans [6]. Latent EBV infection causes EBV-associated malignancies by expressing EBV-encoded transforming proteins and non-coding RNAs. Lytic EBV infection is essential for production of infectious viral particles, enabling virus transmission from cell-to-cell and host-to-host [6]. 

In latently infected cells, the EBV genome is maintained as a nuclear episome that replicates once per cell cycle using the host DNA polymerase [7]. It is usually highly methylated, existing in a repressive chromatin structure. Upon EBV lytic reactivation, EBV lytic genes are expressed in a temporally regulated manner. The first genes transcribed are the viral immediate-early lytic genes, *BZLF1* and *BRLF1*, which encode the transcription factors, Z (Zta) and R (RTA), respectively. Subsequently, Z and R proteins activate both their own and one another’s promoters to greatly amplify their lytic-inducing effects [8].

Anti-herpesvirus agents exert their effects by either directly inhibiting viral enzymes or disrupting the host-virus interactions [9]. These agents can interfere with viral genome replication, interrupt viral protein synthesis, and modulate the host response to viral infection. Acyclovir and ganciclovir are synthetic analogues of guanosine, which are converted into acyclovir phosphate and ganciclovir phosphate by viral and cellular kinases [10]. However, these guanosine analogues have several significant limitations. First, guanosine analogues act on actively replicating viruses and have no effect on latent viruses [11]; and second, acyclovir and ganciclovir therapies can cause symptoms of neurotoxicity, including lethargy, hallucination, seizures, and myelosuppression [12].

Several bioactive compounds in *Cordyceps militaris* (CME) have been identified as cordycepin, cordycepic acids, polysaccharides, macrolides, etc. CME exhibits anti-cancer, anti-depressant, anti-inflammatory, hypoglycemic, anti-microbial, and anti-viral effects due to its diverse bioactive compounds [13]. Cordycepin could exhibit anti-cancer effects, but those effects were sometimes controversial in some points. First, Du et al. showed that cordycepin could enhance EBV gene expression and lytic reactivation [14]. In contrast, we observed that cordycepin significantly suppressed EBV gene expression and lytic reactivation [15]; and secondly, adenosine deaminase (ADA) converts cordycepin to 3′-deoxyinosine which does not exhibit anti-cancer effects as much as the original cordycepin [16,17,18]. T and B cells express ADA on their surfaces because ADA plays an important role in the development of the immune system [19]. The biological activity of cordycepin could be limited by the types of EBV-infected host cells due to its deamination. Thirdly, CME contained a broad spectrum of biological compounds including cordycepin, whose bioactive effects are similar to those of CME [20]. Therefore, we could not exclude that other compounds in CME are implicated in EBV lytic reactivation. 

The molecular mechanisms by which CME and its bioactive compounds exert their anti-viral activities are yet to be determined. With the aim to elucidate this, we first identified adenosine as a major EBV lytic inducing compound in CME and found that a high concentration of adenosine enhances EBV lytic induction in EBVaGC. Second, we found that the adenosine A1 receptor (ADORA1) is required to upregulate *BZLF1,* which encodes for a key regulator (Zta) for the EBV lytic cycle. 

## 2. Results

### 2.1. Cordyceps Extracts Induce EBV Lytic Infection in EBVaGC Cells

Since CMEs have exhibited anti-tumor and anti-viral activities in several tumor cells [15,21], we verified the anti-viral activities in EBV-associated gastric carcinoma (EBVaGC) using SNU719 cells. First, we measured the 50% cytotoxicity (CD_50_) of CME on SNU719 cells after 48 h incubation at 37 °C. The CD_50_ was determined to be 1.1 mg/mL (Figure 1A). Second, we conducted a promoter usage assay to test if CME can affect the usage of EBV promoters, Cp, Qp, and Fp in SNU719 cells treated with CME at 1.1 mg/mL for 48 h. Generally, six EBNA (EBNA1, 2, 3A, 3B, 3C and 4(-LP)) expressions initiate at either the Cp or Wp promoter during type III latency. In type I or II latency, only EBNA1 is expressed by the Qp and Fp involved the lytic cycle [8]. CME increased the frequency of use for the EBV lytic gene promoter, Fp, and suppressed the EBV latent gene promoter, Qp, in SNU719 cells treated with CME at 1.1 mg/mL for 48 h (Figure 1B). Third, we investigated if CME affects the expression of EBV proteins in SNU719 cells. A western blot assay showed that CME induced the expression of EBNA1, Zta, and LMP2A proteins in SNU719 cells treated with CME at 1.1 mg/mL for 48 h (Figure 1C). Similarly, an RT-qPCR assay showed that CME induced the transcription of most EBV genes tested in SNU719 cells treated with CME at 1.1 mg/mL for 48 h (Figure 1D). Finally, we examined if CME affected the production of EBV progeny virus in SNU719 cells. An EBV genome measurement assay showed that CME significantly induced the production of extracellular EBV genome copy numbers in SNU719 cells treated CME at 1.1 mg/mL for 72 h (Figure 1E). Taken together, these results indicate that CME induces EBV lytic reactivation and results in the overproduction of EBV progeny viruses. 

### 2.2. Adenosine is Present in Cordyceps militaris (CME)

Cordycepin (3′-deoxyadenosine) in *C. militaris* has been proposed to be the active component of traditional medications made from this parasitic fungus [22]. Cordycepin is a derivative of the nucleoside adenosine, differing by the absence of oxygen at the 3′ position of its ribose entity. It has been shown previously that the de novo purine nucleotide pathway in *C. militaris* can produce several nucleosides, such as guanosine, inosine, adenosine, and cordycepin [23]. Another study reported that 1,9-dimethylguanine, adenosine, uridine, nicotinamide, 3-methyluracil, 1,7-dimethylxanthine, nudifloric acid, and mannitol were isolated from CME [24]. To confirm these findings, we conducted a high-performance liquid chromatography (HPLC) assay to verify the presence of nucleosides in CME. Our assay showed that CME contains cordycepin, adenosine, guanosine, uridine, and other compounds (Figure 2A). Next, we treated SNU719 cells with these nucleosides and measured their CD_50_ values at 48 h post treatment at 37 °C. We repeated these CD_50_ measurements at least three times for each compound and took the highest CD_50_ values for the next experiments. We expected that the highest CD_50_ concentration might produce the strongest effect in the following several experiments. The final highest CD_50_ values were 375 μM for cordycepin, 820 μM for adenosine, 105 μM for guanosine, and 500 μM for uridine in SNU719 cells (Figure 2B). 

### 2.3. Adenosine Induces EBV Gene Expression

To determine the effects of nucleoside analogues on EBV gene expression, we first evaluated their effects on EBV gene transcription in SNU719 cells. An RT-qPCR assay showed that adenosine induced EBV gene transcriptions most effectively among the nucleoside analogues tested (Figure 3A). All EBV gene transcriptions except *EBNA3C* were upregulated by adenosine, while just one or two genes were upregulated by either cordycepin or guanosine. The transcription of *EBNA3C* was downregulated in SNU719 cells treated with each different nucleoside analogues. *BZLF1* was also significantly upregulated by adenosine or cordycepin similar to CME, but not uridine and guanosine. These results show that adenosine upregulates the transcriptions of most EBV genes, but cordycepin induces only two EBV genes, *EBER1* and *BZLF1*. Second, we evaluated the effects of nucleoside analogues on EBV protein production in SNU719 cells. A western blot assay demonstrated that adenosine significantly enhanced the production of EBV proteins such as EBNA1, LMP2A, and Zta (Figure 3B). Consistent with Figure 3A, cordycepin also slightly enhanced EBV protein production, but this enhancement was weaker than that of adenosine. Third, we evaluated the effects of nucleoside analogues on EBV promoter usage in SNU719 cells. A promoter usage detection assay showed that only adenosine increased the frequency of use for the Fp lytic gene promoter, while it significantly decreased the frequency of use for the Qp latent gene promoter (Figure 3C). Cordycepin and uridine also slightly decreased the frequency of use for the Qp promoter, but there were no differences compared to the control. Taken together, these results indicate that adenosine in CME is one of the effective compounds responsible for inducing EBV lytic gene expression.

### 2.4. Adenosine Induces EBV Lytic Reactivation

The CD_50_ of adenosine on SNU719 cells was defined as 820 μM. Thus, we applied this concentration to our subsequent experiments. However, this concentration is too high to be used as a therapeutic. Thus, we determined the lowest concentration of adenosine to initiate the EBV lytic protein production. To this end, a 2-fold serial dilution of the adenosine CD_50_ concentration was applied to SNU719 cells. After 48 h of incubation, SNU719 cells were harvested and subjected to a western blot assay to determine EBV Zta production. The lowest concentration of adenosine that increased the Zta production in SNU719 cells was <102 μM (Figure 4A). We further tested whether EBV *BZLF1* upregulation by adenosine was reproducible in other EBV-infected epithelial cells such as HEK293-EBV and AGS-EBV. The CD_50_ of adenosine on HEK293-EBV cells was determined to be 780 μM (Appendix AA). HEK293-EBV cells were harvested and subjected to a western blot assay. The lowest concentration of adenosine that was effective in HEK293-EBV cells was <390 μM (Figure 4B). The CD_50_ of adenosine on AGS-EBV cells was determined to be 795 μM (Appendix AB). AGS-EBV cells were harvested and subjected to a western blot assay. The lowest concentration of adenosine that was effective to upregulate Zta expression in AGS-EBV cells was <397 μM (Figure 4C). An immunofluorescence assay (IFA) was used to confirm the lowest concentrations of adenosine that could induce the *BZLF1* upregulation. We applied the IFA on SNU719 cells treated with differing concentrations of adenosine. *BZLF1* induction was observed from 51 μM adenosine (Figure 4D). 

An immunocytochemistry assay (ICA) was also conducted to confirm the *BZLF1* upregulation by lower concentrations of adenosine. We applied the ICA on SNU719 cells treated with differing concentrations of adenosine. *BZLF1* induction was observed from 51 μM adenosine (Appendix A). In regard to treatment time, we observed that *BZLF1* induction was quite strong at 48 h post treatment (Figure 4E). Lastly, we questioned whether adenosine enhances EBV progeny production through *BZLF1* upregulation and measured intracellular and extracellular EBV genome copy numbers to answer this. SNU719 cells were treated with adenosine at 102.5 to 820 μM. After 72 h of incubation, the cells were harvested and subjected to the EBV genome measurement assay. The intracellular EBV genome copy number was increased by adenosine-dose dependently. The extracellular EBV genome copy was significantly increased in a dose dependent manner, 12.5-fold at 410 μM adenosine and 18.7-fold at 820 μM adenosine, respectively (*p* < 0.001) (Figure 4F). Taken together, these results indicate that adenosine leads to EBV lytic reactivation through *BZLF1* induction.

### 2.5. Adenosine Has Strong Affinity to Adenosine A1 Receptor (ADORA1)

To determine the signaling induced by adenosine to induce lytic reactivation, we used virtual docking of adenosine to adenosine receptors. The known major adenosine receptors are adenosine A1 receptor (ADORA1) and adenosine A2A receptor (ADORA2A) [25]. These receptors were subjected to molecular modeling with adenosine and cordycepin using Autodock Vina [26]. Crystal structures of ADORA1 and ADORA2A were obtained from the RCSB PDB protein data bank [27]. 5N2S and 2YDO were ADORA1 and ADORA2A structures, respectively. Both structures were subjected to molecular docking analysis with adenosine or cordycepin (Figure 5). The molecular docking assay showed that 2YDO has stronger affinity to adenosine than cordycepin, while 5N2S had no significant difference in affinity to both compounds. The affinities of adenosine and cordycepin to 2YDO were −7.5 kcal/mol and −7.0 kcal/mol, respectively. The root-mean-square deviation of atomic positions (RMSD) is the measure of the average distance between the atoms of superimposed proteins. The RMSD between an adenosine molecule and an indigenous adenosine in 2YDO was 3.829 Å, while the RMSD between a cordycepin and an indigenous adenosine in 2YDO was 4.926 Å. These results indicate that adenosine virtually binds more competitively to ADORA2A than cordycepin.

### 2.6. Induction of Adenosine A1 Receptor by Adenosine

Since adenosine could be virtually bound to adenosine receptors, the regulation of adenosine receptors by adenosine was analyzed by qRT-PCR and a western blot assay. Adenosine significantly induced the transcription of *ADORA1* and *ADORA2A* (*p* < 0.01, respectively), but not *ADORA2B* (*p* = n.s) (Figure 6A). Compared to the nuclease-free water (NFW) control, *ADORA1* and *ADORA2A* transcription levels were increased 2.9- and 2.5-fold, respectively (*p* < 0.01). Due to the more significant effect of adenosine, we were more interested in its regulation of ADORA1 protein expression. Treatment with adenosine at a series of concentrations enhanced ADORA1 protein expression in SNU719 cells (Figure 6B), HEK293-EBV cells (Figure 6C), and AGS-EBV cells (Figure 6D). Induction of ADORA1 protein expression started at 205 μM in SNU719 cells, 97.9 μM in HEK293-EBV cells, and 198 μM in AGS-EBV cells. These results suggest that adenosine may amplify either *ADORA1* or *ADORA2A* signaling by upregulating these genes. 

### 2.7. Inhibitor of Adenosine A1 Receptor Suppresses EBV Gene Induction

A previous molecular docking study showed that adenosine initiates signaling through adenosine receptors [25]. We postulated that adenosine-mediated EBV lytic reactivation could be blocked by selective adenosine receptor antagonists. We were able to obtain a chemical inhibitor of ADORA1, but not ADORA2A. 8-Cyclopentyl-1,3-dipropylxanthine (DPCPX) is known as a potent and selective antagonist for ADORA1 [28]. We first tested if DPCPX could suppress EBV lytic reactivation induced by adenosine. SNU719 cells were treated with 0.05, 0.5, and 5 μM DPCPX. After 48 h incubation, SNU719 cells were harvested and subjected to western blot assay. We observed that levels of EBNA1 and Zta proteins were significantly less with 5 μM DPCPX treatment along with a decrease in the ADORA1 protein (Figure 7A). We then conducted RT-qPCR to investigate if DPCPX could suppress the induction of *ADORA1* (Figure 7B) and EBV genes by adenosine (Figure 7C). Transcription levels of most EBV genes were increased by adenosine and those of EBV genes were markedly reduced by co-treatment with adenosine and DPCPX. Transcription levels of *ADORA1* and 7 EBV genes were significantly enhanced by adenosine and *LMP1*, *EBER1*, and *EBNA1* were fully reduced by co-treatment with DPCPX and adenosine (*p* = n.s, respectively). Although *LMP2*, *BLLF1*, *BZLF1*, and *BRLF1* did not reduce completely after co-treatment with DPCPX and adenosine, co-treatment with DPCPX and adenosine had the ability to reduce EBV gene transcriptions which were increased by adenosine treatment. Thus, DPCPX could partially inhibit adenosine-mediated EBV gene induction. Taken together, these results suggest that at least ADORA1 signaling is involved in the Zta-mediated EBV lytic reactivation. 

### 2.8. Partial Depletion of ADORA1 Suppresses BZLF1 Induction by Adenosine

Since adenosine was found to induce the transcription of *ADORA1* and *ADORA2A* in SNU719 cells, we hypothesized that adenosine induces EBV lytic reactivation through specific ADORA1 or ADORA2A receptors. To test this hypothesis, we first attempted to specifically deplete *ADORA1* or *ADORA2A* in SNU719 cells using lentiviral shRNA. Three shRNAs per gene were packed into lentiviruses, which were infected into SNU719 cells. Stable cell lines were established by 15 days of selection with puromycin. Only *ADORA1* shRNAs treatment could establish ADORA1-depleted SNU719 cell lines but *ADORA2A* shRNAs treatment could not establish an ADORA2A-depleted SNU719 cell line. We inferred that ADORA2A is required for the survival of SNU719 cells because complete depletion of *ADORA2A* repeatedly caused cell death. Secondly, we validated the *ADORA1*-depleted SNU719 cell lines. RT-qPCR analysis showed that among the three pLKO.1 *ADORA1* shRNAs, TRCN00008038 (*ADORA1*-8038 shR) downregulated *ADORA1* most efficiently (Figure 8A). A western blot analysis showed that the *ADORA1*-8038 shR partially depleted ADORA1 (Figure 8B). Thirdly, we applied adenosine to *ADORA1*-downregulated SNU719 cells (ADORA1-8038 shR) in parallel with control SNU719 cells (CTL shR), followed by western blot and RT-qPCR analyses. The western blot analysis demonstrated that adenosine induced translation of *EBNA1*, *BZLF1*, *EA-D*, and *LMP1* compared to shCTL SNU719 cells (Figure 8C). This induction in control SNU719 cells was suppressed in *ADORA1*-downregulated SNU719 cells. In particular, EBNA1 and Zta protein levels were significantly lower in the *ADORA1*-downregulated SNU719 cells treated with adenosine. Next, we determined transcription level of EBV genes and *ADORA1* using RT-qPCR analysis (Figure 8D). Adenosine significantly enhanced transcriptions of *ADORA1*, *BZLF1*, *BRLF1*, *EBER1*, *EBNA1*, and *LMP1*. However, the transcriptional enhancements of *BZLF1*, *BRLF1*, *EBER, EBNA1*, and *LMP1* were undermined by the depletion of *ADORA1*. Taken together, these results indicate that adenosine can upregulate EBV genes via ADORA1. However, we cannot exclude the possibility that another factor, such as ADORA2A, is involved in initiating EBV lytic reactivation.

### 2.9. Adenosine Increases ATP but Decreases Cyclic Adenosine Monophosphate

ATP is converted via a series of dephosphorylation steps by cytosolic ecto-5′-nucleotidase CD39 and CD73 under extracellular conditions [25]. Adenosine can be transported inside or outside the cell via diffusion or active transport. Adenosine is intracellularly converted to ATP via phosphorylation steps mediated by adenosine kinase and AMP kinase. Thus, adenosine applied to SNU719 cells can be converted to ATP inside SNU719 cells. To determine if adenosine affects ATP levels, the CellTiter-Glo 2.0 assay was conducted to quantitate ATP in SNU719 cells treated with adenosine for 48 h. Before determining the effect of adenosine, we conducted the assay to generate a standard curve for ATP and luminescence (Appendix AA). There was a forward correlation between the amount of rATP and Renilla luciferase activity. As the rATP concentration was increased, the renilla luciferase unit (RLU) was also increased (Appendix AB). Then, we applied adenosine to SNU719 cells at different concentrations and determined both directly corresponding RLUs (Figure 9A) and the relative intracellular ATP amounts, compared to NFW control treatment (Figure 9B). Intracellular ATP production in SNU719 cells was induced by adenosine in a concentration-dependent manner. Until adenosine concentration reached to 205 μM, RLU was increased along with enhancement of the adenosine concentration. Treatment at 205 μM increased intracellular ATP levels by 1.6-fold compared to the NFW control (*p* < 0.001) (Figure 9B). This result indicates that adenosine can be converted to intracellular ATP in a dose-dependent manner. 

ADORA1 suppresses adenylyl cyclase and thus inhibits new production of cAMP. However, ADORA2A activates adenylyl cyclase and consequently enhances cAMP production. Thus, adenosine could activate either ADORA1 or ADORA2A signaling as it is absorbed into SNU719 cells. We thus determined how adenosine receptors are involved in adenosine-mediated EBV lytic reactivation. We conducted the cAMP-Glo^TM^ assay to generate a standard curve for cAMP and luminescence (Appendix AC). There was a reverse correlation between the amount of cAMP and renilla luciferase activity. As the rATP concentration was increased, the renilla luciferase unit (RLU) was decreased (Appendix AD). Similar to the CellTiter-Glo 2.0 assay, we applied adenosine to SNU719 cells in different concentrations and determined both the directly corresponding RLUs (Figure 9C) and the relative intracellular cAMP levels, compared to the NFW control treatment (Figure 9D). As the adenosine concentration was increased, RLU was also increased. This result indicated that intracellular cAMP production in SNU719 cells was suppressed by adenosine in a concentration-dependent manner. Treatment with 820 μM adenosine decreased cAMP levels for 2.8-fold compared to the control (*p* < 0.001) (Figure 9D). These results indicate that adenosine is more likely to initiate ADORA1 signaling rather than ADORA2A signaling. The activation of ADORA1 signaling might play a key role in EBV lytic reactivation through *BZLF1* upregulation. 

### 2.10. Adenosine Suppresses Tumor Development in a Xenograft Mouse Model

To evaluate the anti-tumor effects of adenosine in vivo, we conducted a xenograft model anti-tumor assay. The assay was designed as described previously [21]. Mice were injected subcutaneously with MKN1-EBV cells (5 × 10^6^ cells per mouse) and then adenosine (30 mg/kg) or drinking water were administrated orally every day for three weeks. Tumor growth was determined in all 12 mice and tumor volume was measured every two days until it reached 2000 mm^3^ (Figure 10A). We measured the weights of xenograft mice bearing MKN1-EBV cells-derived tumors and found that the weights were not changed during the whole experimental period (Figure 10B). We then measured the sizes of MKN1-EBV cells-derived tumors on the xenograft mice and found that continuous feeding of adenosine significantly inhibited the growth of MKN1-EBV cells-derived tumors (Figure 10C,D). Mice were sacrificed at 17 days post-feeding. Tumors were removed from mice and were subjected to a western blot assay. As expected, the Zta protein level was clearly increased in tumors from adenosine-fed mice compared to tumors from water-fed mice (Figure 10D). However, EBNA1 and LMP2A protein levels were not changed and were even reduced in tumors from adenosine-fed mice compared to tumors from water-fed mice unexpectedly. These findings indicated that another molecular mechanism was likely to influence the adenosine-mediated downregulation of EBNA1 in xenograft mouse tissues. Taken together, these results suggest that the antiviral effect of adenosine contributes to delayed tumor development in the EBVaGC xenograft mouse.

## 3. Discussion

*C. militaris* is known to contain interesting functional metabolites including various chemical structures and bioactivities. Among *Cordyceps*-derived metabolites, cordycepin is considered to be a potential antitumor drug due to its antitumor activities [29]. Thus, it was reasonable to test whether *Cordyceps* or its functional metabolites have the potential to prevent EBVaGC. To this end, we studied the bioactive effects of CME and its metabolites in EBVaGC. First, we found that CME could induce viral gene expressions which are essential for EBV lytic reactivation; second, we found by HPLC assay that CME contains nucleoside analogues, including adenosine; third, we discovered that adenosine has a strong potential for EBV lytic reactivation in EBVaGC; fourth, we showed that adenosine partially requires ADORA1 signaling to upregulate *BZLF1,* which encodes for a key regulator of EBV lytic cycles; and last, we discovered that *BZLF1* upregulation by adenosine is significantly effective in suppressing EBVaGC development. 

Regarding the *BZLF1* upregulation of cordycepin against EBV infection, there is discord between our previous study [15] and a study by Du et al. [14]. Cordycepin induced EBV lytic reactivation in AGS-EBV cells in Du et al.’s study, while the cordycepin suppressed EBV gene expression in SNU719 cells in our study. However, adenosine treatment of SNU719 and HEK293-EBV cells induced EBV lytic reactivation. One of the reasons for this difference is that the effect caused by cordycepin might be specific to the types of EBV infected cells. SNU719 cells originated from a male patient with EBV-associated gastric carcinoma that was pathologically proven [30], while AGS-EBV was established by infecting AGS cells with EBV isolated from EBV-infected Akata cells from EBV-related Burkitt’s lymphoma [31]. Thus, we expect that original EBV genomes might be epigenetically different from those of gastric carcinoma and Burkitt’s lymphoma. As the EBVaGC genome is known to be hypermethylated compared to other gastric carcinoma cells [32], the EBV genome in EBVaGC is also expected to be more methylated than the EBV genomes in EBV-infected malignancies. The hypermethylation in EBVaGC might contribute to higher resistance to EBV lytic reactivation by cordycepin in SNU719 cells than in other cells. 

Hui et al. reported that EBNA3C expression is important for drug-sensitivity by an HDAC inhibitor and a proteasome inhibitor in LCLs and BL cells [33]. Our study showed that nucleoside analogues reduce the transcription level of EBNA3C in EBVaGC cells (Figure 3A). Moreover, Hui et al. reported that a combination treatment with ganciclovir and an HDAC inhibitor has synergistic effects for anti-EBV in NPC and GC cells [34,35]. Interestingly, their data showed that C666-1 and epithelial cells do not express gp350, late EBV lytic protein, but not Zta expression, early EBV lytic protein. They explained these results as the abortive lytic induction and late lytic proteins are not required for the induction of cell death upon lytic reactivation. Similar to the studies by Hui et al., we cannot detect the gp350 expression upon adenosine and cordycepin. These results might suggest that nucleoside analogues and an HDAC inhibitor/proteasome inhibitor might have a different mechanism for anti-EBV effects and EBV lytic reactivation and depends on cell types and EBV latency types. 

ADORA1 inhibits signal amplification and diminishes cAMP by interacting with the G proteins of the G_I_ and G_0_ family [36]. ADORA2A stimulates signal amplification and increases cAMP by interacting with G_s_, which stimulates adenylyl cyclase [36]. We could establish partially ADORA1-depleted SNU719 cells but not ADORA2A depleted cells. Thus, we hypothesize that functional *ADORA2A* is required for the survival of SNU719 cells. A previous study reported that AMP could prevent the establishment of a latent infection and eradicate latent HSV-1 from the central nervous system [37]. This eradication of latent HSV-1 might be related to HSV-1 lytic reactivation. Adenosine binding to ADORA1 protein decreased intracellular cAMP levels, thus depressing HSV-1 latent infection. Another study reported that cAMP pathways are involved in HSV-1 lytic reactivation [38]. Chlorophenylthio-cAMP, a membrane permeable cAMP analog, was used as a selective activator of cAMP dependent protein kinase (PKA) and inhibited cGMP-dependent phosphodiesterase. Deprivation of chlorophenylthio-cAMP resulted in HSV-1 lytic reactivation in primary neuronal cultures. Our study similarly showed that adenosine treatment to EBVaGC cells upregulates *ADORA1* and initiates an inhibitory cAMP signaling pathway. This inhibition might diminish the cAMP concentration, and eventually induce EBV lytic reactivation through *BZLF1* upregulation in the same manner as the deprivation of Chlorophenylthio-cAMP.

It was interesting that two structurally similar compounds such as cordycepin and adenosine compete to control EBV life cycles. This difference in the bio-activities of the two compounds could be explained by measuring ATP using a CellTiter-Glo 2.0 assay. Adenosine induced ATP production above the control at all concentrations tested (all concentrations, between 51.28 μM and 820 μM), while cordycepin slightly induced ATP production at only low concentrations (23.4 μM, between 23.4 μM and 375 μM, Appendix AE). The ATP induction was undermined at high concentrations of both compounds. Thus, we could infer that low concentrations of adenosine and cordycepin prefer ADORA1 signaling, in which ATP induction was not interrupted by adenylyl cyclase. However, the ADORA1 signaling might be saturated sooner, and later ADORA2A signaling might replace ADORA1 signaling at a high concentration of adenosine.

The difference of bio-activities could also be explained by virtual molecular docking analysis. Both compounds were predicted to dock to ADORA1 and ADORA2A by AutoDock Vina [26]. ADORA1 had no preference in affinity to both compounds but ADORA2A had strong affinity to adenosine. Thus, these two compounds might compete for both adenosine receptors at low concentrations, but adenosine might have a stronger affinity to ADORA2A at a high concentration. The saturation of ADORA1 signaling and preference for adenosine receptors might contribute to the initiation of EBV lytic reactivation through *BZLF1* upregulation in EBVaGC (Figure 11). Our study shows that CME contained both compounds, but it preferentially induced EBV lytic reactivation in EBVaGC. The preference of CME to EBV lytic reactivation might be due to the strong effect of adenosine on adenosine receptors. 

To date, the induction of EBV lytic reactivation has been reported in major cytolytic virus activation therapy for EBV-infected tumors [39,40]. The induction of EBV lytic reactivation enhances the expressions of viral proteins which can trigger the recognition of EBV-infected tumor cells by host immune cells [39,40]. Moreover, lytic reactivation decreases tumor cell growth with EBV infection and induces sensitivity to the cytotoxic effects of antiviral and anti-cancer agents [39,40]. Our previous and current studies demonstrate that adenosine and cordycepin are likely to regulate the EBV life cycle through EBV *BZLF1* upregulation or downregulation. Thus, adenosine and cordycepin can be used to protect host cells at different stages of EBV infection. When EBV establishes strong latent replication and initiates viral oncogenesis, adenosine might be useful to break EBV latent infection and stimulate EBV lytic reactivation. However, when EBV induces drastic lytic replication to proliferate progeny viruses, cordycepin is expected to suppress EBV lytic replication. However, future studies are required to develop cordycepin and adenosine as functional drugs for the prevention of gastric carcinoma. In this study, we clarify molecular mechanisms of adenosine used to induce EBV lytic cycle and produce antiviral activities, raising the possibility that *Cordyceps* can be developed as a medicinal food to prevent EBV infection and EBVaGC development.

## 4. Materials and Methods

### 4.1. Preparation of CMEs

CMEs were provided by MushTech (Chuncheon, Korea). Fresh fruiting bodies for mycelia of *C. militaris* were extracted with 95% ethanol at room temperature (RT) for 3 days. The extracts were filtered, concentrated, sterilized, and dried as previously described [24]. CME was dissolved in nuclease-free dH_2_O to make a stock solution at 8 mg/mL and stored at −20 °C until use. Adenosine, uridine, guanosine, and cordycepin were purchased from Sigma-Aldrich (St. Louis, MO, USA). Adenosine, uridine, and cordycepin were dissolved in nuclease-free dH_2_O. Guanosine was dissolved in DMSO to make a stock solution before use. 

### 4.2. Cell lines and Reagents

Both gastric carcinoma cell lines SNU719 (EBVaGC), AGS (EBV negative gastric carcinoma), and MKN1 (EBV negative gastric carcinoma) were purchased from Korean Cell Line Bank (Seoul, Korea) and cultured in RPMI 1640 (HyClone, GE Healthcare, Pittsburgh, PA, USA) supplemented with 10% fetal bovine serum (HyClone, Marlborough, MA, USA), antibiotics/antimycotics (Gibco, Thermo Fisher Scientific, Waltham, MA, USA), GlutaMAX (Gibco), and 25 mmol/mL HEPES (Sigma-Aldrich) at 37 °C with 5% CO_2_ and 95% humidity. We previously generated AGS-EBV, HEK293-EBV, and MKN1-EBV cells by stably transfecting them with EBV bacmid [15]. AGS-EBV, HEK293-EBV, and MKN1-EBV cells were cultured in DMEM (Hyclone) and supplemented with 10% FBS (HyClone), antibiotics/antimycotics (Gibco), and GlutaMAX (Gibco).

### 4.3. Cytotoxicity Assay

The cytotoxic effects of CME and nucleosides on SNU719 cells were evaluated by a cytotoxicity assay performed using the Cell Counting Kit-8 (CCK-8; Dojindo, Kumamoto, Japan). Briefly, 100 μL of cell suspension (1 × 10^4^ cells/well) was seeded into a 96-well plate. The following day, CME and nucleosides were applied at various concentrations: 0–8000 μg/mL for CME and 1.95–2000 μM for nucleosides. After 48 h post treatment, 10 μL CCK-8 solution was added to each sample. Samples were incubated for another 3 h, and then the absorbance of each cell suspension was measured at 450 nm using an enzyme-linked immunosorbent assay reader. All steps were followed according to the manufacturer’s recommended protocol. 50% cytotoxicity (CD_50_) was approximately determined as previously described [41]. Simply speaking, the middle absorbance first was calculated between the highest absorbance and the lowest absorbance; secondly, the concentration of the compound was evaluated by assigning the middle absorbance to its corresponding concentration of the compound; thirdly, this concentration of compound was named as the CD_50_ concentration. In the following experiments, we treated cells with compounds at CD_50_ concentration for 48 h, then removed old media containing most dead cells, further washed cells with PBS at least two times to remove clearly dead cells, and finally harvested averagely 90% live cells for analysis.

### 4.4. Western Blot Assay

To assess the effects of CME or nucleoside analogues on EBV protein synthesis, western blotting was performed on SNU719 cells treated with 1.1 mg/mL of CME. Treated SNU719 cells were harvested using trypsin at 48 h post-treatment. Cells (10 × 10^6^) were lysed using 100 μL of RIPA lysis buffer (Tris-HCl (50 mM, pH 8.0), NaCl (150 mM), EDTA (2 mM, pH 8.0), 1% NP-40, 0.5% Sodium Deoxycholate, and 0.1% SDS) and supplemented with proteinase inhibitor and phenylmethylsulfonyl fluoride. The cell lysates were further fractionated using the Bioruptorsonicator (Cosmobio, Tokyo, Japan; 5 min, 30 s on/off pulses). Protein in cell lysates was measured using the Bradford assay. Equivalent amounts of protein were separated in 10% sodium dodecyl sulfate polyacrylamide electrophoresis gel and transferred to membranes. Membranes were probed with antibodies against EBV and cellular proteins. EBNA1 (Santa Cruz Biotechnology, Dallas, TX, USA), BZLF1 (Santa Cruz Biotechnology), EA-D (Santa Cruz Biotechnology), LMP1 (Santa Cruz Biotechnology), LMP2A (Santa Cruz Biotechnology), and ADORA1 (Cell Signaling Technology, Danvers, MA, USA) were detected. GAPDH (Cell Signaling Technology) and β-Actin (Santa Cruz Biotechnology) were used as an internal control. Goat-anti-mouse IgG-HRP (Genetex, Irvine, CA, USA) and Goat-anti-rabbit IgG-HRP (Genetex, Irvine, CA, USA were used as secondary antibodies. Antibody-bound proteins were visualized by an enhanced chemiluminescent (ECL) detection reagent (GE Healthcare). Membranes were stripped and reprobed with other antibodies. 

### 4.5. Reverse-Transcriptional Quantitative PCR (RT-qPCR) Assay

RNA was isolated from cells using the RNeasy Mini Kit (Qiagen, Germantown, MD, USA). A total of 2 μg of the purified RNA were subjected to a Superscript II Reverse Transcriptase (Invitrogen, Carlsbad, CA, USA) to synthesize the RNA into cDNA, according to the manufacturers’ protocols. Diluted RT products were analyzed by real-time PCR (LightCycler 96, Roche, Basel, Switzerland). The levels of actin or GAPDH in each sample were used as the internal control for each RT-qPCR assay. RT-qPCR with RNAs without reverse transcription were conducted to serve as a negative control in each reaction. Primers specific for adenosine receptors such as *ADORA1* (F: 5′-ACC TGG AGG TCT TCT ACC TAA TC-3′, R: 5′-TCT TCA GCT CCT TCC CAT AGT-3′), *ADORA2A* (F: 5′-CTT GGG TTC TGA GGA AGC AG-3′, R: 5′-CAG CAG CTC CTG AAC CCT AG-3′), and *ADORA2B* (F: 5′-ATC TCC AGG TAT CTT CTC-3′, R: 5′-GTT GGC ATA ATC CAC ACA G-3′) were used [42]. Sequences of the other primer sets used in this study can be provided upon request. 

### 4.6. Promoter Usage Assay

RNA was isolated from SNU719 cells using an RNeasy Mini Kit (Qiagen, Germantown, MA, USA) Then, purified RNAs were synthesized into cDNA by using Superscript II Reverse Transcriptase (Invitrogen, Carlsbad, CA, USA). Using the prepared cDNA, EBV promoter usage affected by CME and nucleosides was examined by performing conventional PCR. Primer sequences for *Actin*, EBV Qp, EBV Cp/Wp, and EBV Fp were as previously published. cDNA was amplified in a 25 μL reaction solution containing 5 μL of 5× reaction mix, 5 μL of 5× TuneUp solution, 1 μL of Taq-plus polymerase, and 2.5 μL of 10 pmol forward/reverse primer. The following cycle conditions were used: 95 °C for 3 min; 30 cycles of 95 °C for 10 s, 55 °C for 30 s, and 72 °C for 30 s; followed by 72 °C for 10 min. The reactions were performed using a TaKaRa PCR Thermal Cycler (TaKaRa, Kyoto, Japan) and then run on a 1.2% agarose/TBE gel. 

### 4.7. EBV Genome Measurement Assay

To investigate the intracellular EBV genome copy number, SNU719 cells were seeded in 6-cm plates and treated with of CME (1.1 mg/mL) or adenosine on following day. After 48 h, SNU719 cells were lysed and sonicated using a Bioruptorsonicator (Cosmobio, Tokyo, Japan; 5 min, 30 s on/off pulses). Then, total genomic DNA (gDNA) was isolated as previously described [15]. The resultant gDNA (50 ng) was subjected to qPCR analysis. The relative intracellular EBV genome copy number was determined as compared to internal controls such as *Actin* or *GADPH*. To determine the extracellular EBV copy number, SNU719 cells were seeded in 150-mm plates and treated with CME (1.1 mg/mL) or adenosine (0, 102.5, 205, 410, and 820 μM) on the following day. After 72 h, 20 mL culture medium samples of SNU719 cells were collected from the plates. The culture samples were filtered through a 0.45-μm syringe filter, loaded onto a 20% sucrose cushion in PBS, and subjected to ultracentrifugation (CP100WX, Hitachi, Japan) at 27,000 rpm for 90 min. The virus pellet treated with 0.5 units DNase I (M059S, Enzynomics, Daejeon, Korea) for 30 min at 37 °C and then incubate for 10 min at 75 °C to inactivate the enzyme activity. The samples were lysed in 25 μL FA lysis buffer (EDTA (1 mM, pH 8.0), HEPES-KOH (50 mM, pH 7.5), and NaCl (140 mM)) and 250 μL RIPA buffer (Tris-HCl (50 mM, pH 8.0), NaCl (150 mM), EDTA (2 mM, pH 8.0), 1% NP-40, 0.5% Sodium Deoxycholate, and 0.1% SDS), sonicated using a Bioruptorsonicator for 5 min (30 s on/off pulses), and incubated 5 μL proteinase K (20 mg/mL, Promega, Madison, WI, USA) for 2 h at 50 °C. The virus pellet mixture was subjected to phenol-chloroform extraction and ethanol precipitation to extract the extracellular EBV genomes. Finally, the extracellular EBV genomes were resuspended in 100 μL nuclease-free dH_2_O and qPCR analysis was used to quantify viral DNAs by using primer sets specific for EBV *EBNA1* or *BZLF1*. Relative extracellular EBV genome copy number was determined compared to an internal negative control (i.e., no treatment). For EBV lytic reactivation, 3 mM NaB and 20 ng/mL TPA were treated as positive control in EBV genome measurement assay as mentioned previously [14]. 

### 4.8. High-Performance Liquid Chromatography (HPLC) Assay 

To determine the nucleosides present in CME, an HPLC was conducted as described previously [24].

### 4.9. Immunofluorescence (IFA) and Immunocytochemistry (ICA) Assays

In order to conduct an IFA, SNU719 cells were grown on coverslips in 24-well plates and treated with 820 μM adenosine for 0, 24, 36, and 48 h. Treated cells were fixed with 4% paraformaldehyde for 20 min and were permeabilized with 0.25% Triton X-100 in PBS for 15 min. Treated cells were blocked using 1% BSA in PBS containing 0.1% Tween 20 for 30 min. Samples were stained with the BZLF1 antibody (1:40). After overnight incubation at 4 °C, coverslips were washed 3× in PBS and treated with Alexa-488 (Thermo Fisher Scientific, Waltham, MA, USA) for 1 h at RT. Alexa-488 was used to detect BZLF1 after the BZLF1 antibody was bound to BZLF1. After washing 3× in PBT (PBS containing 0.5% Triton X-100), coverslips were mounted with DAPI (SouthernBiotech, Birmingham, AL, USA). Samples were analyzed using immunofluorescence confocal microscopy. 

In order to conduct ICA, 1.5 × 10^5^ and 3.0 × 10^5^ SNU719 cells were seeded on a cover slip loaded in each well of a 24-well plate for control and adenosine treatments, respectively. SNU719 cells were treated with adenosine in a series of concentrations (51–820 μM). Then, these SNU719 cells were fixed and permeabilized in 100% methanol for 60 min at −20 °C. SNU719 cells were washed twice with PBS followed by once with 1 × PBS/0.1% Tween-20 and blocked with 1% BSA and 0.2% skim milk for 60 min at room temperature. Then, 3% H_2_O_2_ was added to block endogenous peroxidase activity. SNU719 cells were stained with an anti-Zta monoclonal antibody (#sc-53904, Santa Cruz Biotechnology) diluted 1:50 in 1 × PBS/0.1% Tween-20 for overnight at −4 °C and further incubated with secondary antibody (goat-a-mouse-HRP, Jackson Immuno Research, West Grove, PA, USA) diluted 1:50 for 30 min at room temperature. DAB substrate (DAKO, K3468; diluted 1 drop/mL as per manufacturer’s instructions) was added for 5 min to detect positive dark spots. A cover slip containing stained SNU719 cells was taken from 24-well plate and laid on a slide glass. A nuclei counterstain was done on the cover slip using DAPI Fluoromount-G^®^ (SouthernBiotech, Birmingham, AL, USA) to confirm the absolute cell number. Samples were analyzed using immunofluorescence microscopy. 

### 4.10. Lentiviral Transduction

pLKO.1 vector-based shRNA constructs for adenosine receptors (TRCN00008036, TRCN00008037, TRCN00008038 for ADORA1) were purchased from Sigma-Aldrich. Control shRNAs (shCTL) were generated in the pLKO.1 vector with the target sequence 5′-TTATCGCGCATATCACGCG-3′. Lentiviruses were produced using envelope and packaging vectors pMD2.G and pSPAX2 as described previously [43]. SNU719 cells were infected with lentivirus stocks carrying pLKO.1-puro vectors by overlaying the lentivirus stock on SNU719 cells for 24 h. Then, the lentivirus stocks were replaced with fresh RPMI medium and later treated with 1.0 μg/mL puromycin at 48 h post-infection. RPMI medium with 1.0 μg/mL puromycin was replaced every 2 to 3 days. Cells were selected with puromycin at least for 14 days and then subjected to further analyses.

### 4.11. ATP Assay

To analyze intracellular ATP levels, we used the CellTiter-Glo 2.0 reagent (Promega) according to the manufacturer’s protocol. Cells at 2.0 × 10^6^ were seeded in 6-well plates and treated with 51.25, 205.00, and 820.00 μM of adenosine or 23.44, 93.75, and 375.00 μM of cordycepin for 48 h, respectively. Treated cells were trypsinized, centrifuged, and resuspended in 1.0 × 10^4^ per 100 μL medium. CellTiter-Glo 2.0 reagent was added to each sample in 1:1 volume and mixed on a shaker for 2 min, and then incubated at RT for 10 min in the dark. The luminescence was measured using a TECAN infinite M200PRO. An ATP standard curve from 10 nM to 1 μM was also determined using ribonucleotide triphosphates (rATP, Promega).

### 4.12. cAMP Assay 

To analyze intracellular cAMP, we used the cAMP-Glo Assay (Promega) according to the manufacturer’s protocol. First, 2.0 × 10^6^ cells were seeded in 6-well plates and were treated with 12.81, 51.25, 205.00, and 820.00 μM adenosine for 48 h, respectively. Treated cells were trypsinized, centrifuged, and resuspended in 2.0 × 10^4^/100 μL DPBS. After centrifuging the sample to remove DPBS, an aliquot of 20 μL of cAMP Glo lysis buffer was added to each pelleted sample and mixed by vortexing for 1 min, and then incubated at RT for 15 min. Then, 40 μL of cAMP Glo detection solution was add to each sample and mixed on a shaker for 1 min, and then incubated at RT for 20 min. Then, 80 μL of Kinase Glo Reagent was added to each sample and mixed on a shaker for 1 min, and then incubated at RT for 10 min in the dark. The luminescence was measured using a TECAN infinite M200PRO. A cAMP standard curve from 0 nM to 7.81 nM was determined according to the manufacturer’s protocol.

### 4.13. Docking Assay and Binding Score 

To determine the highest binding affinities of nucleosides to ADORA1 or ADORA2A, virtual molecular docking of adenosine and cordycepin to 5N2S (ADORA1 crystal structure) or 2YDO (ADORA2A crystal structure) was performed using AutoDock (version 4.2.6, The Scripps Research Institute, La Jolla, CA, USA) and AutoDock Vina (version 1.1.2, https://sourceforge.net/projects/autodock-vina-1-1-2-64-bit/). Crystal structures of 5N2S and 2YDO were converted to PDBQT formats using AutoDock Tools (ADT), version 1.5.6. Then, Kollman united atom partial charges were assigned for the receptors. The grid size for 5N2S search space was set at 44 Å × 36 Å × 36 Å, centered on the binding pocket of 5N2S, with a default grid point spacing of 1.000 Å. However, the grid sizes for the 2YDO search space were set at −23.91 Å × 20.15 Å × −25.37 Å, centered on the binding pocket of 2YDO, with a default grid point spacing of 1.000 Å. The Lamarckian genetic algorithm was used with a population size of 10 dockings and energy evaluations. Results were clustered according to the RMSD criterion.

### 4.14. Ethics Statement

Animal experiments were conducted in accordance with the National Research Council’s Guide (IACUC, Seoul, Korea) for the Care and Use of Laboratory Animals. The experimental protocol was approved by the Animal Experiments Committee of Duksung Women’s University (permit number: 2014-016-007, 1 September 2014).

### 4.15. Xenograft model Antitumor Assay

NOD/SCID mice (female, 5-week-old; Raonbio Co., Ltd., Seoul, Korea) were used as xenograft animal models. Mice were individually accommodated in a pathogen-free controlled environment (23–27 °C under a 12-h day/12-h night cycle) and provided food and water ad libitum. To produce tumors, mice were first divided into 2 groups (*n* = 12). The animals in one group were subcutaneously implanted with 5 × 10^6^ human EBVaGC, MKN1-EBV cells into the dorsum next to the right hind leg. After 14 days, both groups were subdivided into 2 subgroups (*n* = 6, respectively) and orally administrated drinking water and adenosine (30 mg/kg) for 17 days. Administration amounts of adenosine were determined based on our previous studies; 30 mg/kg for quercetin and isoliquiritigenin [44]. Tumors were identified and measured every other day using a standard caliper; tumor size was calculated using [tumor length (mm) × tumor width (mm)^2^]/2 as previously described [21]. After the tumor size had reached 2000 mm^3^, animals were euthanized, and tumors were harvested.

### 4.16. Statistical Analysis

The *p*-values were calculated by 2-tailed Student’s *t*-test using Excel (Microsoft, Redmond, WA, USA).

## Figures and Tables

**Figure 1 ijms-20-01286-f001:**
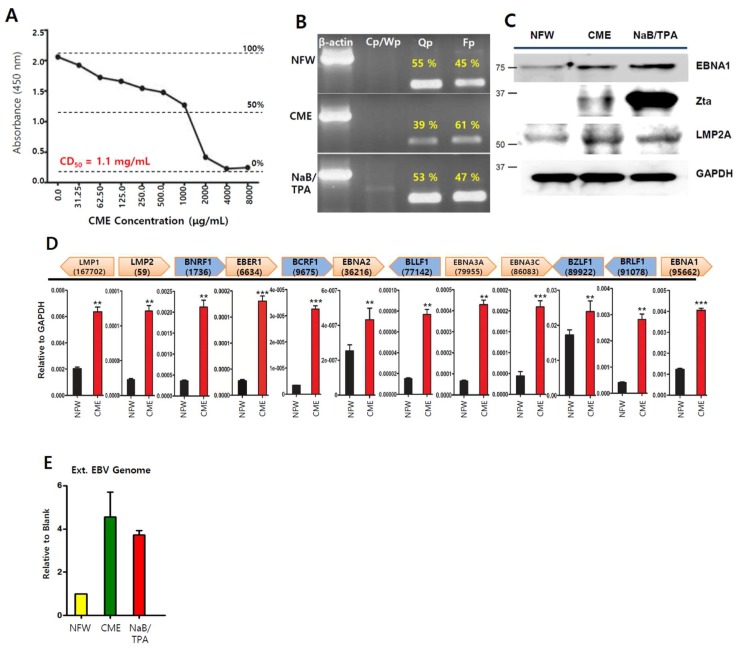
Anti-viral effects of extracts of *Cordyceps militaris* (CMEs) on Epstein–Barr virus (EBV) gene expression and DNA replication. (**A**). Measurement of cytotoxicity (CD_50_)of CME on SNU719 cells using a cytotoxicity assay, CME was applied to SNU719 cells at 1.1 mg/mL as follows; (**B**) Determination of CME effect on EBV promoter activity. SNU719 was treated with CME at 1.1 mg/mL for 48 h and then RNA was isolated and synthesized cDNA. Band densities of Qp and Fp were measured using imageJ program and are represented as percentages. NaB/TPA were used as positive control; (**C**) Determination of CME effect on EBV protein expression; (**D**) Determination of CME effect on EBV gene transcriptions using RT-qPCR; (**E**) Determination of CME effect on EBV progeny production. CME and NFW = *Cordyceps militaris* extracts and nuclease-free water, respectively. ** *p* < 0.01, *** *p* < 0.001 (Student’s *t*-test).

**Figure 2 ijms-20-01286-f002:**
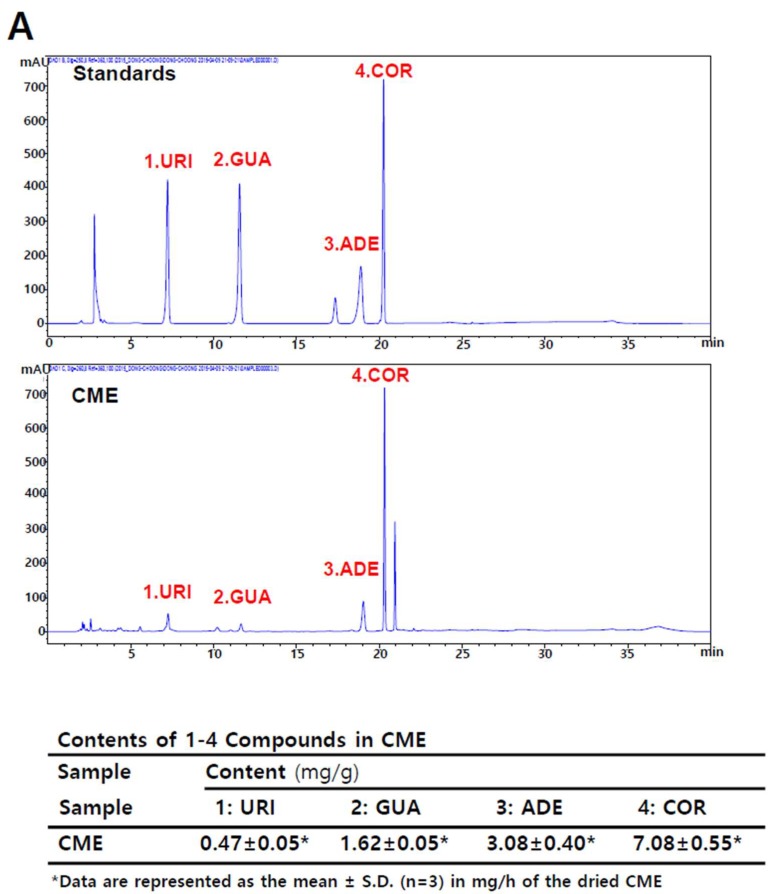
Identification of nucleoside analogues in CME. (**A**) Investigation of nucleoside analogues of CME by a high-performance liquid chromatography (HPLC) profile and reference EBV genome; (**B**) Measurement of CD_50_ of nucleoside analogues on SNU719 cells. ADE, COR, URI, and GUA = adenosine, cordycepin, uridine, and guanosine, respectively.

**Figure 3 ijms-20-01286-f003:**
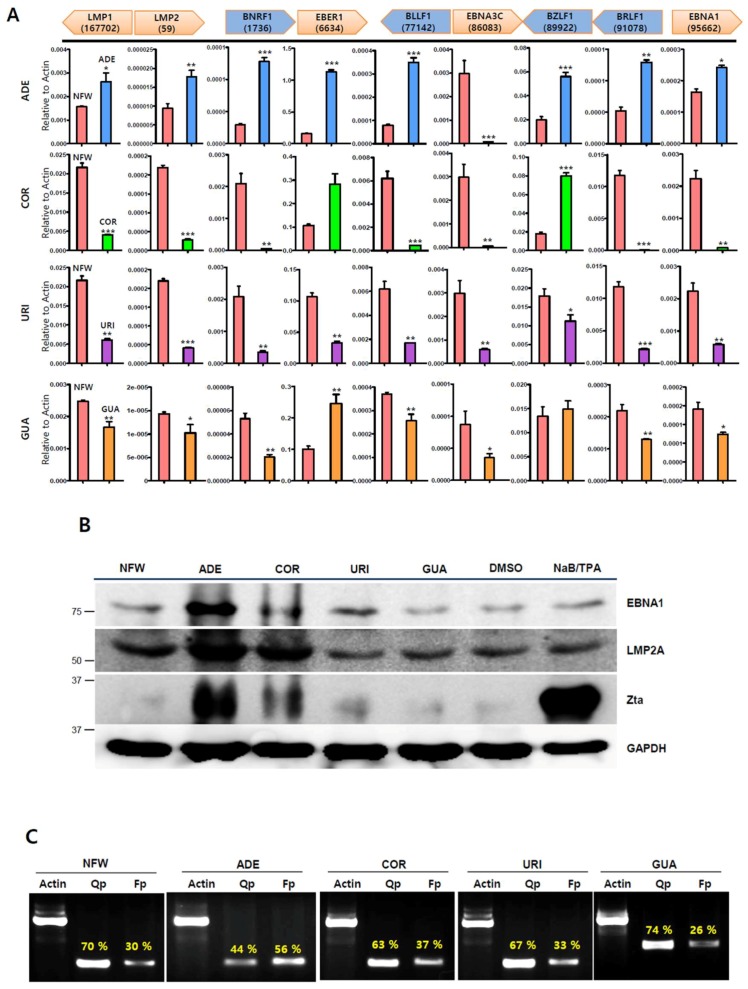
Anti-viral effects of nucleoside analogues on EBV gene expression. SNU719 cells were treated with nucleoside analogues at their CD_50_ concentrations. (**A**) Determination of nucleoside analogues effects on EBV gene transcriptions. CD_50_ concentration of each nucleoside analogues treated SNU719 cells for 48 h; (**B**) Determination of nucleoside analogues effects on EBV protein expression. Those CD_50_ of nucleoside analogues treated SNU719 cells for 48 h, EBV protein expressions were analyzed by a western blot assay; (**C**) Determination of nucleoside analogues effects on EBV promoter activity. SNU719 cells were treated with CD_50_ concentration of nucleoside analogues for 48 h and purified RNA to synthesize cDNA. ADE, COR, URI, GUA, and NFW = adenosine, cordycepin, uridine, guanosine, and nuclease-free water, respectively. CD_50_s of ADE, COR, URI, and GUA were 820 μM, 375 μM, 500 μM, and 105 μM, respectively. * *p* < 0.05, ** *p* < 0.01, *** *p* < 0.001 (Student’s *t*-test).

**Figure 4 ijms-20-01286-f004:**
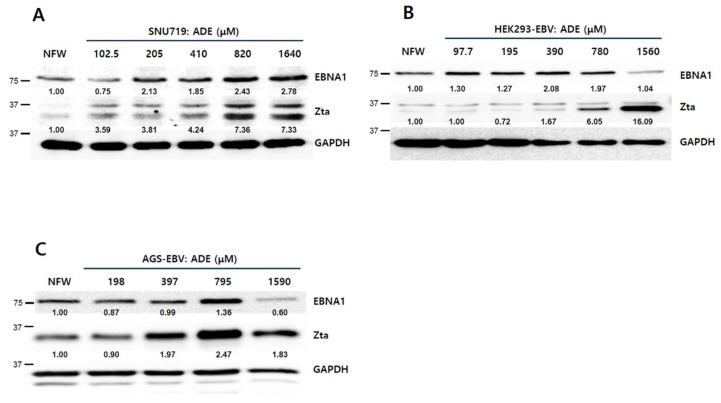
Anti-viral effects of adenosine on EBV lytic reactivation. (**A**) Determination adenosine-does dependent EBV lytic reactivation in SNU719 cells. Adenosine was applied in a series of concentrations, relative to its CD_50_ against SNU719 cells (820 μM); (**B**) Determination of adenosine-dose dependent EBV lytic reactivation in HEK293-EBV cells. Adenosine was applied in a series of concentrations, relative to its CD_50_ against HEK293-EBV cells (780 μM); (**C**) Determination adenosine –dose dependent EBV lytic reactivation in AGS-EBV cells. Adenosine was applied in a series of concentrations, relative to its CD_50_ against AGS-EBV cells (795 μM); (**D**) Visualization of adenosine-derived Zta protein induction in a concentration-dependent manner. SNU719 cells were treated with 51.25, 205, and 820 μM adenosine for 48 h. Cells were stained with anti-BZLF1 (green) and nuclei were stained with DAPI (blue); (**E**) IFA-based confirmation of adenosine-derived Zta protein induction in a time-dependent manner. SNU719 cells were treated with 820 μM adenosine for 24 h, 36 h, and 48 h; (**F**) Determination of the effects of adenosine on EBV progeny production. SNU719 cells were treated with 0, 102.5, 205, 410, and 820 μM adenosine for 72 h and then measured intracellular and extracellular EBV genome copy number. GAPDH was used as loading control. ADE and NFW = adenosine and nuclease-free water, respectively. * *p* < 0.05, ** *p* < 0.01, *** *p* < 0.001 (Student’s *t*-test).

**Figure 5 ijms-20-01286-f005:**
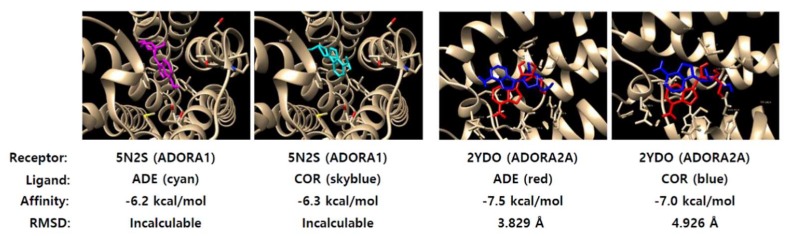
Virtual docking of nucleosides to adenosine A1 receptor (ADORA1) and ADORA2A. Docking of adenosine and cordycepin to ADORA1 and ADORA2A. 5N2S and 2YDO are partial crystal structures of ADORA1 and ADORA2A. ADE and COR = adenosine and cordycepin, respectively.

**Figure 6 ijms-20-01286-f006:**
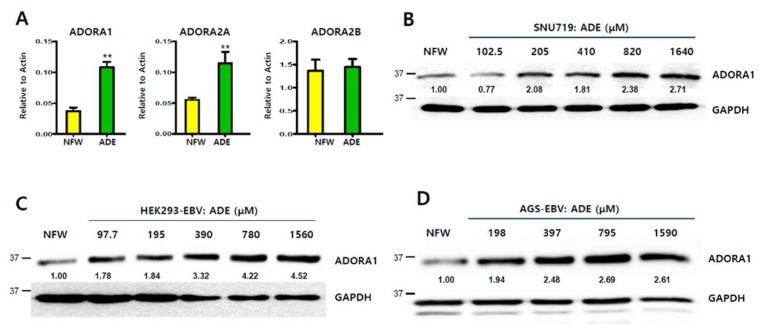
Induction of adenosine receptors by adenosine. (**A**) Induction of transcription of adenosine receptor genes by adenosine in SNU719 cells treated with 820 μM adenosine for 48 h; (**B**) Induction of ADORA1 protein expression in SNU719 cells treated with a series of concentrations of adenosine for 48 h; (**C**) Induction of ADORA1 protein expression in HEK293-EBV cells treated a series of concentrations of adenosine for 48 h; (**D**) Induction of ADORA1 protein expression in AGS-EBV cells treated with a series of concentrations of adenosine for 48 h. GAPDH was used as loading control. ADE and NFW = adenosine and nuclease-free water, respectively. ** *p* < 0.01 (Student’s *t*-test).

**Figure 7 ijms-20-01286-f007:**
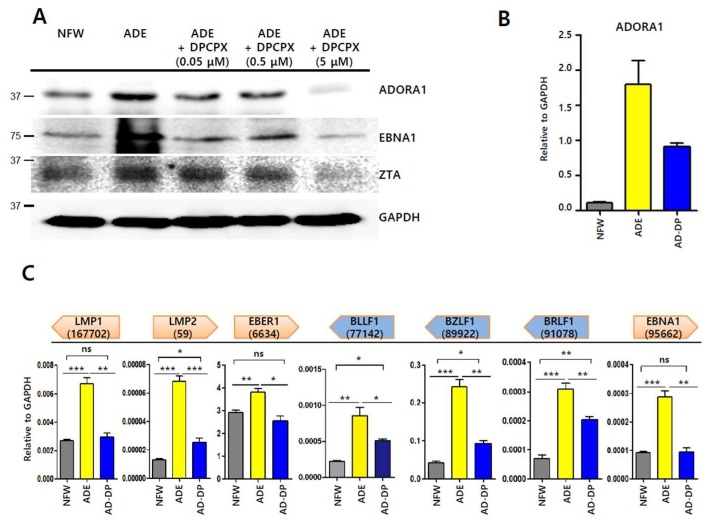
Inhibitor-based investigation of EBV lytic reactivation by adenosine. (**A**) Inhibition of adenosine-derived EBV protein induction by 8-cyclopentyl-1,3-dipropylxanthine (DPCPX), an adenosine inhibitor. SNU719 cells were co-treated with 820 μM adenosine and a series of concentrations of DPCPX for 48 h; (**B**) Confirmation of ADORA1 transcription in SNU719 cells treated with 820 μM adenosine or 5 μM DPCPX for 48 h; (**C**) Inhibition of adenosine-derived EBV gene transcriptional induction by DPCPX. SNU719 cells were co-treated with 820 μM adenosine and 5 μM DPCPX for 48 h. ADE and NFW = adenosine and nuclease-free water, respectively. * *p* < 0.05, ** *p* < 0.01, *** *p* < 0.001 (Student’s *t*-test), ns: not significant.

**Figure 8 ijms-20-01286-f008:**
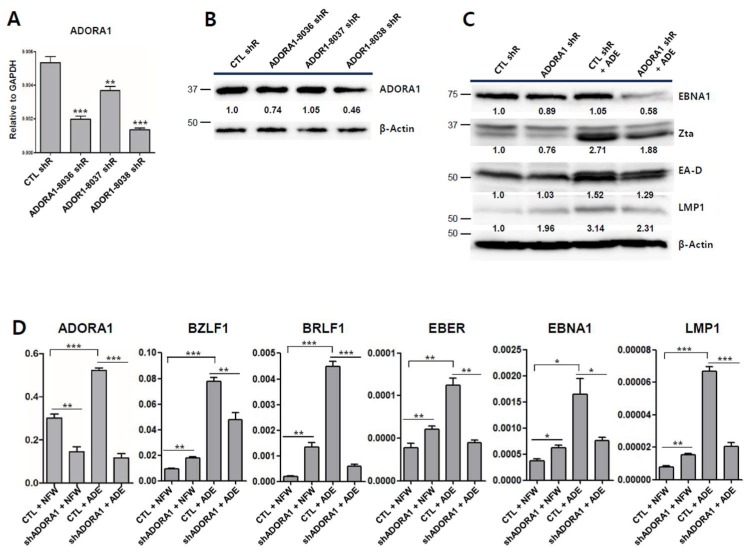
shRNA-based investigation of EBV lytic reactivation by adenosine. (**A**) Confirmation of *ADORA1* transcription in ADORA1 knock-downed SNU719 cells; (**B**) Confirmation of knock-down efficiency for the ADORA1 protein in ADORA1 knock-downed SNU719 cells using western blotting; (**C**) Effects of adenosine on EBV gene translation in control and partially *ADORA1* depleted SNU719 cells. Both of the SNU719 cells were treated with 820 μM adenosine for 48 h. β–actin was used as loading control; (**D**) Effects of adenosine on EBV gene translation in partially *ADORA1* depleted SNU719 cells. Control and partially *ADORA1* depleted SNU719 cells were treated with 820 μM adenosine for 48 h. ADE and NFW = adenosine and nuclease-free water, respectively. * *p* < 0.05, ** *p* < 0.01, *** *p* < 0.001 (Student’s *t*-test).

**Figure 9 ijms-20-01286-f009:**
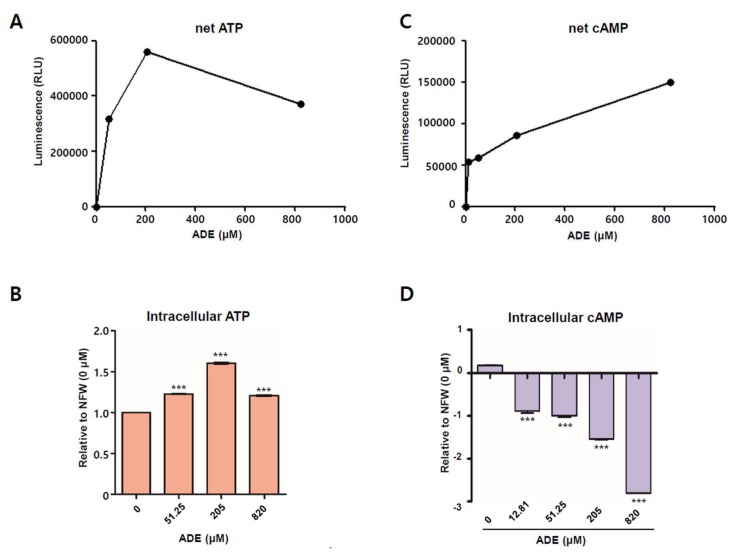
Measurement of intracellular ATP and intracellular cAMP induction by adenosine. (**A**) Measurement of net ATP induction by adenosine in SNU719 cells. Net ATP levels were represented as RLU directly relative to the adenosine concentration; (**B**) Measurement of ATP induction by adenosine in SNU719 cells. SNU719 cells were treated with 51.25, 205.00, and 820.00 μM of adenosine for 48 h. ATP levels are represented relative to the NFW control (0 μM); (**C**) Measurement of net cAMP induction by adenosine in SNU719 cells. Net cAMP levels were represented as RLU directly relative to the adenosine concentration; (**D**) Measurement of cAMP induction by adenosine in SNU719 cells. SNU719 cells were treated with 12.81, 51.25, 205.00, and 820.00 μM of adenosine for 48 h. cAMP levels are represented relative to the NFW control (0 μM). ADE and NFW = adenosine and nuclease-free water, respectively. *** *p* < 0.001 (Student’s *t*-test).

**Figure 10 ijms-20-01286-f010:**
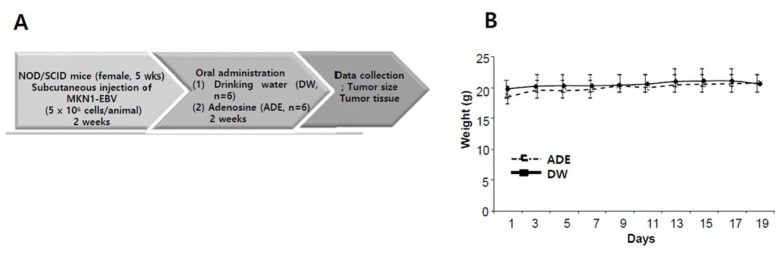
Anti-tumor effect of adenosine (ADE) in a xenograft NOD/SCID mouse bearing EBVaGC (MKN1-EBV). (**A**) A schematic diagram of the xenograft mouse model antitumor assay. DW group (*n* = 6) and ADE (30 mg/kg) group (*n* = 6); (**B**) Weight measurement of mice bearing MKN1-EBV cell-derived tumors during a whole experimental period; (**C**) Representative image of a NOD/SCID mouse bearing MKN1-EBV cells-derived tumors; (**D**) Determination of the antitumor effects of adenosine on the development of MKN1-EBV cells-derived tumors in NOD/SCID mice. Image of xenograft mouse model after three weeks from oral administration; (**E**) Confirmation of adenosine-derived *BZLF1* induction in MKN1-EBV cells-derived tumors in NOD/SCID mice. ADE and DW = adenosine and drinking water, respectively.

**Figure 11 ijms-20-01286-f011:**
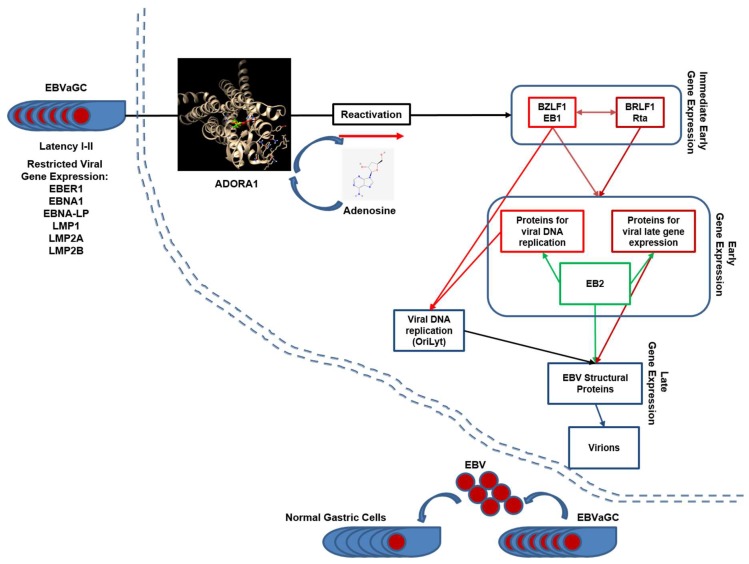
A working model of adenosine regulation of EBV gene expression in SNU719 cells (adapted from Centre International de Recherche en Infectionogie). Adenosine functions as a transcriptional activator of EBV genes.

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
