# Peer review of "Adenosine Induces EBV Lytic Reactivation through ADORA1 in EBV-Associated Gastric Carcinoma"

_ijms, 2019, doi:10.3390/ijms20061286_

Round 1

Reviewer 1 Report

Most of the careless mistakes have been corrected in the revised manuscript. However, the following points should be discussed and clarified before acceptance for publication:

1. Author response #4. Gp350 should work well in different EBV positive epithelial cells. Actually, gp350 were not expressed suggests that adenosine induces early phase, but not the late phase of EBV lytic cycle in SNU719 cells. Such abortive lytic induction is similar to that observed in some cell lines like C6661 and SNU719. Please refer to “Int J Cancer. 2016 Jan 1;138(1):125-36. doi: 10.1002/ijc.29698” and “Int J Cancer. 2012 Oct 15;131(8):1930-40. doi: 10.1002/ijc.27439. Epub 2012 Mar 8” for more information. The authors should cite and discuss these 2 papers in their manuscript to describe the abortive lytic induction in the SNU719 cells. In fact, late lytic proteins are not required for the induction of cell death upon lytic reactivation.

2. Author response #10. The authors should discuss the possible implication of upregulation of EBNA3C. It plays an important role in the survival of EBV-positive B cell lines as suggested by many previous papers and a recent paper “Oncotarget. 2018 May 18;9(38):25101-25114. doi: 10.18632/oncotarget.25341”. What would be the biological consequences of upregulation of EBNA3C in the gastric carcinoma cells?

3. Author response #22. The manuscript did not show the data of combining adenosine induction with antiviral drugs such as ganciclovir. The authors should instead discuss the direct effect of lytic induction on the killing of EBV-positive gastic carcinoma cell lines. For instance, EBV lytic induction by histone deacetylase inhibitors such as SAHA or romidepsin was shown to induce direct killing of EBV-positive epithelial cells (Int J Cancer. 2010 May 15;126(10):2479-89. doi: 10.1002/ijc.24945) (Int J Cancer. 2012 Oct 15;131(8):1930-40. doi: 10.1002/ijc.27439. Epub 2012 Mar 8). The authors should discuss these papers regarding the direct killing effects upon lytic cycle reactivation in epithelial cells.

Author Response

1) Author response #4. Gp350 should work well in different EBV positive epithelial cells. Actually, gp350 were not expressed suggests that adenosine induces early phase, but not the late phase of EBV lytic cycle in SNU719 cells. Such abortive lytic induction is similar to that observed in some cell lines like C6661 and SNU719. Please refer to “Int J Cancer. 2016 Jan 1;138(1):125-36. doi: 10.1002/ijc.29698” and “Int J Cancer. 2012 Oct 15;131(8):1930-40. doi: 10.1002/ijc.27439. Epub 2012 Mar 8” for more information. The authors should cite and discuss these 2 papers in their manuscript to describe the abortive lytic induction in the SNU719 cells. In fact, late lytic proteins are not required for the induction of cell death upon lytic reactivation.

Thank you for your comment. We revised and added the Discussion section in line 338-348.

2) Author response #10. The authors should discuss the possible implication of upregulation of EBNA3C. It plays an important role in the survival of EBV-positive B cell lines as suggested by many previous papers and a recent paper “Oncotarget. 2018 May 18;9(38):25101-25114. doi: 10.18632/oncotarget.25341”. What would be the biological consequences of upregulation of EBNA3C in the gastric carcinoma cells?

Thank you for your comment. Hui et al. (2018) reported that EBNA3C related to apoptotic cell death by HDAC inhibitors in B cells and EBNA3C expressing BL cells became more susceptible to the apoptotic cell death by HDAC inhibitor/proteasome inhibitor. In this respect, HDAC inhibitor/proteasome inhibitor and nucleoside analog have different mechanisms to induce apoptotic cell death and EBV lytic reactivation in EBV-infected malignancy. Because EBVaGC cells belong to EBV latency type I and EBNA3A expression is restricted. Our data support that nucleoside analogs decrease transcription level of EBNA3C (Fig. 3A). So, EBNA3C is not major target for nucleoside analog to induce EBV lytic reactivation in EBVaGC cells. We added the sentence at line 338-348.

3) Author response #22. The manuscript did not show the data of combining adenosine induction with antiviral drugs such as ganciclovir. The authors should instead discuss the direct effect of lytic induction on the killing of EBV-positive gastic carcinoma cell lines. For instance, EBV lytic induction by histone deacetylase inhibitors such as SAHA or romidepsin was shown to induce direct killing of EBV-positive epithelial cells (Int J Cancer. 2010 May 15;126(10):2479-89. doi: 10.1002/ijc.24945) (Int J Cancer. 2012 Oct 15;131(8):1930-40. doi: 10.1002/ijc.27439. Epub 2012 Mar 8). The authors should discuss these papers regarding the direct killing effects upon lytic cycle reactivation in epithelial cells.

Thank you for your comment. Of course, other pharmacological compounds such as HDAC inhibitor, proteasome inhibitor, and DNMT inhibitor, effects EBV lytic reactivation and anti-EBV. In this paper suggest the potential compound of other nucleoside analogs except ganciclovir which is analogue of 2’-deoxy-guanosine. In this regard, we confirmed CME induce EBV gene expression and EBV lytic reactivation (Fig. 1) and CME has several nucleoside analogues such adenosine, cordycepin, and etc (Fig. 2). Among 4 types of nucleoside analogues, only adenosine induce EBV gene expression and EBV lytic reactivation (Fig. 3). So, our paper could emphasize that adenosine has the ability for anti-EBV as novel nucleoside analogue in EBVaGC cells.  

Reviewer 2 Report

Choi and others present here that adenosine treatment leads to viral lytic state in EBV-associated gastric carcinoma cell lines. The authors seem to have problems in language, and the text must be edited. The authors have paid significant amount of effort on this manuscript, but some more efforts are needed for publication, as mentioned below.

Fig. 1 and 3; why the effects of the extract or nucleoside analogs were monitored at the concentrations of CD50? It means 50% of the cells are dead. Too strong, isn’t it? The effects that the authors observe might just be caused by the toxicity, but not by the direct effects of the substances.

Page 3 line 110-112; “The Qp promoter initiates … EBV lytic genes.”

It seems the authors do not have sufficient knowledge on EBV, or they have a problem in English writing. Why the “genes” should be plural? Are there two or three EBNA1 genes or more? Does the Qp/Fp drive any other latent or lytic genes besides EBNA1?

Fig. 1B, 3C; the authors need to do these experiments by qRT-PCR. Otherwise, these data should be eliminated because they are unreliable. For example, in Fig. 1B, why CME treatment resulted in decreased levels of both of the Qp and Fp?  

Fig. 1D, 3A, 7C; the EBV genes and promoters are oddly aligned. Does Cp or Wp drive BNRF1 or EBER1 gene expression? Which gene does Qp drive? Also, this reviewer would put EBNA1 after BRLF1.

Fig. 3A and B; why did COR treatment cause LMP2A protein upregulation when mRNA level was markedly decreased.

Fig. 4A why was LMP2A protein decreased by 820 uM ADE while the same treatment notably induced LMP2A in the Fig. 3B? Showing such data would just degrade quality of their manuscript.

Author Response

1) Fig. 1 and 3; why the effects of the extract or nucleoside analogs were monitored at the concentrations of CD50? It means 50% of the cells are dead. Too strong, isn’t it? The effects that the authors observe might just be caused by the toxicity, but not by the direct effects of the substances.

 Thank you for your comment. We used CD50 concentration of each nucleoside analogs and cordycepin to detect EBV lytic reactivation in EBVaGC cells. Although the high concentration cause toxicity for cells, Uridine and Guanosine cannot induce EBV lytic reactivation (Fig. 3A and 3B). The CD50 concentration of adenosine have strong EBV lytic reactivation (Fig. 3A and 3B) and these effects depends on dose-dependent manner (Fig. 4). These results suggest that the toxicity according for the high concentration is not induce EBV lytic reactivation and adenosine is unique compounds to induce EBV lytic replication among nucleoside analogs.    

 2) Page 3 line 110-112; “The Qp promoter initiates … EBV lytic genes.” It seems the authors do not have sufficient knowledge on EBV, or they have a problem in English writing. Why the “genes” should be plural? Are there two or three EBNA1 genes or more? Does the Qp/Fp drive any other latent or lytic genes besides EBNA1?

We are sorry to out mistake. We revised the sentence at line 89-92.

Generally, six EBNAs (EBNA1, 2, 3A, 3B, 3C and 4(-LP)) expressions initiate at either Cp or Wp promoter during type III latency. In type I or II latency, only EBNA1 is expressed by Qp and Fp involved the lytic cycle

 3) Fig. 1B, 3C; the authors need to do these experiments by qRT-PCR. Otherwise, these data should be eliminated because they are unreliable. For example, in Fig. 1B, why CME treatment resulted in decreased levels of both of the Qp and Fp?

Thank you for your comment. The usage of Qp was reduced but those of Fp was increased by CME treatment. Consequently, CME increase the transcription levels of all EBV genes (Fig. 1D) and slightly upregulate the expression of EBV lytic protein, Zta (Fig. 1C). These data mean that CME has effect for EBV lytic replication in EBVaGC cells.

 4) Fig. 1D, 3A, 7C; the EBV genes and promoters are oddly aligned. Does Cp or Wp drive BNRF1 or EBER1 gene expression? Which gene does Qp drive? Also, this reviewer would put EBNA1 after BRLF1.

We are sorry to our mistake. We removed promoter and rearranged figure graph according for comment (Fig. 1D, 3A, 7C). 

 5) Fig. 3A and B; why did COR treatment cause LMP2A protein upregulation when mRNA level was markedly decreased.

We reported what we observed the LMP2A protein upregulation, but sorry to say that we could explain why this protein upregulation occurs in spite of the LMP2A mRNA downregulation.

 6) Fig. 4A why was LMP2A protein decreased by 820 uM ADE while the same treatment notably induced LMP2A in the Fig. 3B? Showing such data would just degrade quality of their manuscript.

We accepted this comment and deleted LMP2A data from Fig. 4A.

Reviewer 3 Report

This manuscript still need the incorporate the changes suggested by other reviewers.

Author Response

1) This manuscript still need the incorporate the changes suggested by other reviewers.

Thank you for your comment. We revised the manuscript as other reviewer’s comment.

Round 2

Reviewer 2 Report

This reviewer think it is acceptable now.

This manuscript is a resubmission of an earlier submission. The following is a list of the peer review reports and author responses from that submission.

Round 1

Reviewer 1 Report

Comments:

1. Substantial correction in English is needed.

2. INTRODUCTION is insufficient to explain the reason for conducting this study.

3. DISCUSSION is insufficient to explain the reason for the opposing results obtained in this study and from that by Du et al.

4. Most of the western blot results were blurry and unclear.

5. The author did not explain why EBV lytic reactivation could reduce tumor growth.

6. The concentration for adenosine to be effective in inducing EBV lytic cycle is too high for it to develop into a drug-able compound.

7. The last paragraph of the manuscript clearly revealed that the authors have inadequate understanding on the background of EBV life cycle.

Major corrections:

1. INTRODUCTION, in what sense does EBVaGC constituents the largest group of EBV-associated malignancies, can the author provide some numerical data to support this claim. Also, in reference 3, EBVaGC incidence rate is 16-18% in the US and Germany; while 4.3% in China, it is inappropriate to conclude that "EBVaGC occupies more than 10% in total gastric cancer cases".

2. RESULT 1, the author should be more careful in reporting the concentration of CD50 in the main text, the legend as well as within the figure. For instance, in the main text, the CD50 is 1.1 mg/μL, while in Figure 1A, the concentration iis 1.1mg/mL, and in the legend, the concentration is 1.1μg/mL.

3. The promoter usage assay is insufficient to report the activation of the EBV promoters; the author should use other assay methods such as luciferase reporter assay.

4. Measuring external EBV genome copy number is insufficient to report the production of EBV progeny, the author should also test for the expression of EBV late lytic proteins such as VCA-p18 and gp350, as well as perform EBV transfection assay to support "CME induces EBV lytic reactivation and results in overproduction of EBV progeny viruses".

5. Can the author explain why EBV latent proteins EBNA 1 and LMP2A are expressed when Qp is suppressed as shown in Figure 1B to D and in Figure 3C?

6. All assays should be subjected to statistical analysis.

7. The information in all the figure legends is insufficient. The author should add more information such as concentration and duration of compound treatment etc.

8. CD50 measurements were repeated at least 3 times for all compounds according to the author, however, the concentration of the compound treated were not the same for each time as the author mentioned that they took the highest CD50 values for the next experiments. This could not be considered as a triplicate when the concentration was different each time.

9. Can the author explain how they could draw the speculation in line 125-126?

10. Line 137, EBNA3 was downregulated by adenosine, so it is inappropriate to say "nine genes tested were upregulated by adenosine"

11. Can the author explain how they determined the lowest concentration od adenosine for ZTA induction is <102 μM for SNU719, <780 μM for HEK293-EBV and <397 μM for AGS-EBV cells. How can the author draw such a conclusion when the Zta expression can be detected at 102, 780 and 397 μM adenosine, respectively? As stated by the author, could Zta be detected in these three cells treated with 1 μM of adenosine?

12. Did the author notice there were no increase in Zta expression from 51 μM to 820 μM in Figure 4D? Is it possible that the Zta expression observed was due to spontaneous lytic of the cells?

13. The author claimed that the BZLF1 induction was "quite strong" at 48h post treatment (Figure 4E). However, there is only a few signal observed in this panel, the author should provide a percentage of Zta signals observed in each of the conditions.

14. For the docking assay, a validation assay such as crystallography, thermal shift assay or other means should be perform to validate the result of the virtual binding.

15. Line 233, can the author explain how they determined the concentration for inducing the expression of ADORA1 protein was 102.5 μM for SNU719 when the intensity of the band from the NFW sample was stronger than that treated with 102.5 μM. 

16. Can the author explain why "the transcription levels of ADORA1, LMP1, LMP2, BZLF1 and EBNA1 were greatly enhanced by adenosine, yet they were weakly reduced by DPCPX" when the levels of induction were similar to the levels of reduction for all these genes? Most of the time, both the levels of induction and reduction were around 3 fold.

17. Can the author explain why “transcription of EBER1, BLLF1 and BRLF1 were not affected by DPCPX"? For EBER1, the gene is not even responsive to adenosine treatment, not to mention the effect of adenosine + DPCPX; for BLLF1, the gene induction was around 4 fold while the reduction was around 3 fold, so it is somewhat responsive to DPCPX treatment.

18. For both points 16 and 17, the author should provide statistical analysis in support to wordings such as "greatly", "weakly".

19. Figure 8B, the knockdown of ADOR1 was very subtle, the author should repeat this assay.

20. Figure 9A and C, the author should repeat the standard curve, the two standard curve were not appealing. Moreover, the standard curve should be reported as triplicate.

21. Can the author restate the purpose of this study? Why was it reasonable to test whether cordyceps has the potential to prevent EBVaGC?

22. Can the author explain why the upregulation of BZLF1 by adenosine is significantly effective in suppressing EBVaGC development, given that lytic reactivation has been shown to promote tumorigenesis?

23. The author should provide literature report to support their claim on line 389, "the EBV genome in EBVaGC is also expected to be more methylated than the EBV genomes in Burkitt lymphoma or EBV-negative gastric carcinoma cells."

24. The author should explain why "When EBV establishes strong latent replication, it is recommended to initiate EBV lytic reactivation by adenosine." (Line 434-435)

Minor corrections:

1. "Western blot assay" either capitalized all or all in small letter

2. Please remove all the red correction lines in Figure 1C and E; Figure 3B and C; Figure 4A, B and C; Figure 5; Figure 7C; Figure 8A; Figure 9A and C; Figure 11

3. Difficult to understand line 18 to 19.

4. Line 21, either "However, another study reported that...." or " However, other studies reported that"

5. Line 26, ".....upregulating BXLF1 that encodes for a key...."

6. Repeated meaning in ABSTRACT line 23-25, point 1 and 2.

7. Difficult to understand "causes largely" in line 43.

8. Line 45, either "enabling virus to transmit from cell to cell and host to host" or” enabling virus transmission from cell-to-cell and host-to-host"

9. Line 53, "The EBV lytic reactivation occurs only in EBV-positive cells" this is a fact that does not need to restate.

10. Line 55, "effects"

11. Line 56, "interactions"

12. Line 64, remove "and" before "etc"

13. Difficult to understand, line 66 " Cordycepin could exhibit anticancer effects as expected, but whose effects were sometimes controversial in some points", why is it "as expected" for Cordycepin to exhibit anticancer effects?

14. Line 70, add "the" after "as much as"

15. Line 72, add "the" after "the development of"

16. Line 73, "the biological activity of Cordycepin could be limited by the types pf EBV......"

17. Line 73, whose deamination, could the author state the subject specifically.

18. Difficult to understand line 73-76

19. Line 79, wrong usage of semi-colon

20. Line 81, "upregulates BZLF1 which encodes for a key...."

21. Line 88, "CME can affect the usage of EBV promoters Cp, Qp and Fp in SNU719 cells"

22. Line 92, "Third, we investigated if CME affects the expression..."

23. Difficult to understand, line 104, "as follows" follows what?

24. Line 113, "differing by the absence of oxygen at the 3" position..."

25. Line 141, "and Cordycepin similar to the upregulation by CME"

26. Line 150, "but these decrease were very slight", can the author to use another word to replace " very slight", "very" is a vague term.

27. Line 164, wrong usage of semi-colon.

28. Line 166, "we determined the lowest concentration of adenosine capable of increasing EBV lytic protein production", can the author use another term to replace "increasing", "increasing" here does not have a meaning.

29. Line 187, "up to 250-500-fold..." the author can directly write "approximately 500 fold" as supported by Figure 4F.

30. Line 190, the author can directly write ADORA1 involved instead of "an adenosine receptor"

31. Line 234, "These results suggested that"

32. Line 245, the author should name the "previously identified adenosine receptors" directly here.

33. Line 249-250, the author should name the concentration of DPCPX used instead of saying "ten-fold serial dilutions of DPCPX".

34. Line 257, "Thus, DPCPX could partially inhibit"

35. Line 277, "TRCN00008038 (ADORA1-8038 shR) downregulated ADORA1 most efficiently"

36. Difficult to understand, line 285 "RT-qPCR analysis determined what effect adenosine did cause on EBV gene transcription"

37. Difficult to understand in lines 308-309, 331, 334-335.

38. Line 337, "decreased cAMP levels for 2.8 fold..."

39. Line 337, "This result indicated that adenosine was more likely to initiate..."

40. Line 339, "a key role in EBV lytic reactivation through BZLF1 upregulation" remove "the"

41. Line 343, “The assay was designed as described previously..."

42. Line 345, “drinking water were administered orally..."

43. Line 347, “measured the weights..."

44. Line 347, "they" refers to?

45. Line 348, “were not changed during the whole experimental period"; "We then measured the sizes..."

46. Line 351, "Tumors were removed from mice and were subjected to..."

47. Difficult to understand, line 353-355, "However..."

48. Line 355, can the author replace "work for” to another word.

Author Response

Major corrections:

1. INTRODUCTION, in what sense does EBVaGC constituents the largest group of EBV-associated malignancies, can the author provide some numerical data to support this claim. Also, in reference 3, EBVaGC incidence rate is 16-18% in the US and Germany; while 4.3% in China, it is inappropriate to conclude that "EBVaGC occupies more than 10% in total gastric cancer cases".

Thanks for your advice. We revised and marked the numerical data and added the reference paper at line 41-45.

Khan et al reported that 1.8% of all cancer deaths represented EBV associated malignancies cases in 2010 worldwide and the largest mortality from EBV associated malignancies was for EBVaGC from 1990 to 2010 (69,081 cases) (REF1). EBVaGC accounts for averagely 10% in all gastric cancer cases and the incidence of EBVaGC depends on geographic distribution and environmental factors (REF1, 2)[3].

REF 1: Khan G, Hashim MJ. Global burden of deaths from Epstein-Barr virus attributable malignancies 1990-2010. Infect Agent Cancer. 2014;9(1):38. Published 2014 Nov 17. doi:10.1186/1750-9378-9-38

REF 2: Jácome, Alexandre Andrade dos Anjos, et al. "Epstein-Barr virus-positive gastric cancer: a distinct molecular subtype of the disease?." Revista da Sociedade Brasileira de Medicina Tropical 49.2 (2016): 150-157.

2. RESULT 1, the author should be more careful in reporting the concentration of CD50 in the main text, the legend as well as within the figure. For instance, in the main text, the CD50 is 1.1 mg/μL, while in Figure 1A, the concentration iis 1.1mg/mL, and in the legend, the concentration is 1.1μg/mL.

We are very sorry for our mistake. We checked and revised the usage of CME concentration 1.1 mg/mL as CD50 value.

3. The promoter usage assay is insufficient to report the activation of the EBV promoters; the author should use other assay methods such as luciferase reporter assay.

Thank for your mention. We are sorry for our mistake. We revised the word ‘activation of the EBV promoters’ to ‘the frequency of use of EBV promoters’. Our promoter usage assay is measurement tool for the frequency of use of EBV promoters such as Cp, Qp, and Fp. We used the same experiments generally and published paper (REF).

REF: Lee, M., Son, M., Ryu, E., Shin, Y. S., Kim, J. G., Kang, B. W., ... & Kang, H. (2015). Quercetin-induced apoptosis prevents EBV infection. Oncotarget, 6(14), 12603.

4. Measuring external EBV genome copy number is insufficient to report the production of EBV progeny, the author should also test for the expression of EBV late lytic proteins such as VCA-p18 and gp350, as well as perform EBV transfection assay to support "CME induces EBV lytic reactivation and results in overproduction of EBV progeny viruses".

Thanks for your suggestion. We already tried gp350 antibody to test using western blot assay and immuno-fluorescence assay, but gp350 antibody was not working in SNU719 cells. So, we used Zta antibody to show the EBV lytic reactivation in Fig.1C. In case of EBV lytic reactivation, all EBV gene transcripts increased, we measured EBV gene transcription levels to support our hypothesis.

5. Can the author explain why EBV latent proteins EBNA 1 and LMP2A are expressed when Qp is suppressed as shown in Figure 1B to D and in Figure 3C?

In EBV lytic reactivation, all genes of EBV (>80 genes) were increased. EBNA1 and LMP2A also were increased in lytic state.

6. All assays should be subjected to statistical analysis.

Fixed. We added statistical analysis and marked in all figures and all figure legends.

7. The information in all the figure legends is insufficient. The author should add more information such as concentration and duration of compound treatment etc.

Many thanks for your advice. We marked concentration and incubation time in all figure legends, respectively.

8. CD50 measurements were repeated at least 3 times for all compounds according to the author, however, the concentration of the compound treated were not the same for each time as the author mentioned that they took the highest CD50 values for the next experiments. This could not be considered as a triplicate when the concentration was different each time.

We are very sorry for our mistake. We revised CME concentration 1.1 mg/mL in all experiments.

9. Can the author explain how they could draw the speculation in line 125-126?

We are sorry for our mistake. We deleted the sentence.

10. Line 137, EBNA3 was downregulated by adenosine, so it is inappropriate to say "nine genes tested were upregulated by adenosine"

We are sorry for our mistake. We checked data and revised “All EBV genes except EBNA3C” and add the sentence “The transcription of EBNA3C was downregulated in SNU719 cells treated with each 4 types of different nucleoside analogues”.

11. Can the author explain how they determined the lowest concentration od adenosine for ZTA induction is <102 μM for SNU719, <780 μM for HEK293-EBV and <397 μM for AGS-EBV cells. How can the author draw such a conclusion when the Zta expression can be detected at 102, 780 and 397 μM adenosine, respectively? As stated by the author, could Zta be detected in these three cells treated with 1 μM of adenosine?

To analyze the adenosine-dependent EBV lytic reactivation, we choose CD50 concentration of adenosine from 3 different EBV-gastric carcinoma cell lines. We measured cytotoxicity to get CD50 concentration from 3 different EBV-gastric carcinoma cells and then those of cells were treated with serial concentration of adenosine for 48 h. We added cytotoxicity results for HEK293-EBV and AGS-EBV cells (Supplement Fig. 1).   

12. Did the author notice there were no increase in Zta expression from 51 μM to 820 μM in Figure 4D? Is it possible that the Zta expression observed was due to spontaneous lytic of the cells?

As shown Fig. 4A, we already tested serial concentration of adenosine treatment in SNU719 cells and determined at low concentration of adenosine treatment also can induce Zta expression slightly. Consistent with Fig 4A, immuno-fluorescence data showed that a high concentration of adenosine treatment groups have partially increased Zta dots and density. Also, we replaced Fig. 4F to support Figure 4 data.

As seen in Supplemental Figure 2, we additionally conducted immunocytochemistry assay (ICA) to reconfirm that lower concentration of adenosine could induce Zta expression. Anti-Zta antibody was detected in SUN719 cells using DAB substrate kit which could react with HRP conjugated in Goat anti-mouse IgG antibody. Zta expression was getting stronger as we treated SNU719 cells with higher concentrations of adenosine. This result was likely to support the BZLF1 induction by lower concentrations of adenosine presented in Fig 4D. 

13. The author claimed that the BZLF1 induction was "quite strong" at 48h post treatment (Figure 4E). However, there is only a few signal observed in this panel, the author should provide a percentage of Zta signals observed in each of the conditions.

We tried immuno-fluorescence assay more than 3 times. It was very difficult to visualize the BZLF1 from SNU719 treated with 820 μM adenosine. The high concentration of adenosine can cell death and the attachments of those cells were decreased. Those cells might be detached during immuno-fluorescence assay. So we undergone BZLF1 visualization using immuno-fluorescence assay. To support Figure 4 results, we measured intracellular and extracellular EBV copy number from SNU719 cells treated with 0, 102.5, 205, 410, and 820 μM adenosine for 72h.

14. For the docking assay, a validation assay such as crystallography, thermal shift assay or other means should be perform to validate the result of the virtual binding.

It is really true to conduct several validation experiments to validate our virtual binding, but these experiments are beyond subjects directly related to our manuscript.

15. Line 233, can the author explain how they determined the concentration for inducing the expression of ADORA1 protein was 102.5 μM for SNU719 when the intensity of the band from the NFW sample was stronger than that treated with 102.5 μM. 

We are sorry for our mistake. We checked data and revised line in 203-204. ADORA1 induction started at 205 μM adenosine in SNU719 cells. This result was related with our Fig.4F. Extracellular EBV copy number also increased from 205 μM adenosine in SNU719 cells.

16. Can the author explain why "the transcription levels of ADORA1, LMP1, LMP2, BZLF1 and EBNA1 were greatly enhanced by adenosine, yet they were weakly reduced by DPCPX" when the levels of induction were similar to the levels of reduction for all these genes? Most of the time, both the levels of induction and reduction were around 3 fold.

Although mRNA level of ADORA1 showed 5 μM DPCPX cannot fully reduce ADORA1 transcriptional level, Figure 7A showed 5 μM DPCPX treatment can completely reduce adenosine-derived ADORA1 expression more than NFW treatment groups. So, enhanced mRNA levels such as LMP1, LMP2, BZLF1 and EBNA1 were decreased completely by co-treatment of 5 μM DPCPX.

We fixed and added the sentence “Transcription levels of ADORA1 and seven types of EBV gene were significantly enhanced by adenosine and LMP1, EBER1, and EBNA1 were fully reduced by co-treatment with DPCPX and adenosine (p=n.s). Although transcription levels of LMP2, BLLF1, BZLF1, and BRLF1 did not reduced completely by co-treatment with DPCPX and adenosine, co-treatment with DPCPX and adenosine has the ability to reduce EBV gene transcriptions which were increased by adenosine treatment” at line 217-222.

17. Can the author explain why “transcription of EBER1, BLLF1 and BRLF1 were not affected by DPCPX"? For EBER1, the gene is not even responsive to adenosine treatment, not to mention the effect of adenosine + DPCPX; for BLLF1, the gene induction was around 4 fold while the reduction was around 3 fold, so it is somewhat responsive to DPCPX treatment.

Thank you for your advice. We checked statistical analysis to support our data. Statistical analysis showed that co-treatment with adenosine and DPCPX can reduce EBV gene transcription levels which are increased by adenosine treatment. EBER transcriptional level was fully reduced by DPCPX and BZLF1 and BRLF1 also partially decreased by DPCPX. We fixed and added the sentence “Transcription levels of ADORA1 and seven types of EBV gene were significantly enhanced by adenosine and LMP1, EBER1, and EBNA1 were fully reduced by co-treatment with DPCPX and adenosine (p=n.s). Although transcription levels of LMP2, BLLF1, BZLF1, and BRLF1 did not reduced completely by co-treatment with DPCPX and adenosine, co-treatment with DPCPX and adenosine has the ability to reduce EBV gene transcriptions which were increased by adenosine treatment” at line 217-222.

18. For both points 16 and 17, the author should provide statistical analysis in support to wordings such as "greatly", "weakly".

Thank you for your suggestion. We added statistical analysis all data.

19. Figure 8B, the knockdown of ADOR1 was very subtle, the author should repeat this assay.

To support the data, we performed western blot assay and replaced Fig. 8B. We added transcriptional level of ADORA1 in Fig. 8D and marked statistical analysis.

20. Figure 9A and C, the author should repeat the standard curve, the two standard curve were not appealing. Moreover, the standard curve should be reported as triplicate.

Thank you for your suggestion. We rearranged Fig. 9. ATP and cAMP standard curves were tested as triplicate. We attached table to support triplicate in supplemental figure 2F. Also, we added statistical analysis in Fig.9A and 9C.

21. Can the author restate the purpose of this study? Why was it reasonable to test whether cordyceps has the potential to prevent EBVaGC?

Cordyceps contain pharmacological compounds and those compounds have various effects for anti-microbial, anti-inflammatory and anti-cancer. Therefore, our aim for study was that cordyceps have property for anti-EBVaGC and which compounds are related with the EBV lytic reactivation and anti-EBVaGC. Fig. 1 shown that CME have the property for the EBV lytic reactivation.

Next, we investigated which compounds have effects for EBV lytic reactivation in EBVaGC. Most of all, we focused a cordycepin, because this compounds is one of the important compounds in cordyceps and is known as 3’-deoxyadenosine or a derivative of the nucleoside adenosine. As anti-herpesvirus agents, acyclovir and ganciclovir, are synthetic analogues of guanosine, we tested whether other nucleoside analogues, cordycepin and adenosine, have properties for anti-EBVaGC. Our HPLC data shown that CME contains adenosine, uridine, and cordycepin. Also, our data shown that adenosine has more effective than cordycepin which known as adenosine analogue in EBVaGC cells. Our study presented that CME has effects for the EBV lytic reactivation and this effects are related with adenosine rather than cordycepin in EBVaGC.

22. Can the author explain why the upregulation of BZLF1 by adenosine is significantly effective in suppressing EBVaGC development, given that lytic reactivation has been shown to promote tumorigenesis?

We added the sentence “To date, induction of EBV lytic reactivation has been reported the major cytolytic virus activation therapy for EBV-infected tumors. The induction of EBV lytic reactivation enhances the expressions of viral protein which can trigger recognition of EBV-infected tumor cells by host immune cells. Moreover, lytic reactivation decrease tumor cell growth with EBV infection and induce sensitivity to the cytotoxic effects of antiviral and anticancer agents.” at line 369-373.

REF 3: Novalic, Z., et al. "Agents and approaches for lytic induction therapy of Epstein-Barr Virus associated malignancies." Medicinal Chemistry (Los Angeles) 6 (2016): 449-466.

REF 4: Wildeman, Maarten A., et al. "Cytolytic Virus Activation Therapy for Epstein-Barr Virus–Driven Tumors." Clinical Cancer Research (2012).

23. The author should provide literature report to support their claim on line 389, "the EBV genome in EBVaGC is also expected to be more methylated than the EBV genomes in Burkitt lymphoma or EBV-negative gastric carcinoma cells."

Thank you for your advice. We have a mistake. We marked reference paper ‘Comprehensive molecular characterization of gastric adenocarcinoma’. This paper reported that EBV infected tumors had a higher prevalence of DNA hypermethylation than any cancers reported by TCGA. We deleted “in Burkitt’s lymphoma” in line 333-334.

24. The author should explain why "When EBV establishes strong latent replication, it is recommended to initiate EBV lytic reactivation by adenosine." (Line 434-435)

We wanted to say that adenosine is useful to reactivate EBV lytic infection from its latent infection. The latent infection is known to largely inducing the viral oncogenesis. This we revised the above sentence into “When EBV establishes strong latent replication and initiates viral oncogenesis, adenosine might be useful to break EBV latent infection and stimulate EBV lytic reactivation”.

Minor corrections:

1. "Western blot assay" either capitalized all or all in small letter

We are sorry for our mistake. We revised letter.

2. Please remove all the red correction lines in Figure 1C and E; Figure 3B and C; Figure 4A, B and C; Figure 5; Figure 7C; Figure 8A; Figure 9A and C; Figure 11

Thank for your advice. We fixed.

3. Difficult to understand line 18 to 19.

We revised the sentence ‘Cordyceps species are known to contain numerous bioactive compounds, including cordycepin.’

4. Line 21, either "However, another study reported that...." or " However, other studies reported that"

Fixed.

5. Line 26, ".....upregulating BXLF1 that encodes for a key...."

Fixed.

6. Repeated meaning in ABSTRACT line 23-25, point 1 and 2.

Fixed.

7. Difficult to understand "causes largely" in line 43.

Fixed. Delete ‘largely’ and ‘all’.

8. Line 45, either "enabling virus to transmit from cell to cell and host to host" or” enabling virus transmission from cell-to-cell and host-to-host"

Fixed.

9. Line 53, "The EBV lytic reactivation occurs only in EBV-positive cells" this is a fact that does not need to restate.

Thank for your mention. We deleted the sentence.

10. Line 55, "effects"

Fixed.

11. Line 56, "interactions"

Fixed.

12. Line 64, remove "and" before "etc"

We removed ‘and’.

13. Difficult to understand, line 66 " Cordycepin could exhibit anticancer effects as expected, but whose effects were sometimes controversial in some points", why is it "as expected" for Cordycepin to exhibit anticancer effects?

We deleted ‘as expected’.

14. Line 70, add "the" after "as much as"

Fixed.

15. Line 72, add "the" after "the development of"

Fixed.

16. Line 73, "the biological activity of Cordycepin could be limited by the types pf EBV......"

Fixed.

17. Line 73, whose deamination, could the author state the subject specifically.

Fixed.

18. Difficult to understand line 73-76

We revised the sentence from ‘CME contained a broad spectrum of biological compounds except cordycepin whose bioactive effects are similar to those by CME’ to ‘CME contained a broad spectrum of biological compounds including cordycepin whose bioactive effects are similar to those by CME’

19. Line 79, wrong usage of semi-colon

Fixed.

20. Line 81, "upregulates BZLF1 which encodes for a key...."

Fixed.

21. Line 88, "CME can affect the usage of EBV promoters Cp, Qp and Fp in SNU719 cells"

Fixed.

22. Line 92, "Third, we investigated if CME affects the expression..."

Fixed.

23. Difficult to understand, line 104, "as follows" follows what?

We can not find ‘as follows’

24. Line 113, "differing by the absence of oxygen at the 3" position..."

Fixed.

25. Line 141, "and Cordycepin similar to the upregulation by CME"

Fixed. We revised the sentence “BZLF1 was also significantly upregulated by adenosine or cordycepin similar to CME, but not uridine and guanosine.”

26. Line 150, "but these decrease were very slight", can the author to use another word to replace "very slight", "very" is a vague term.

Fixed. We revised the sentence “but these decreases were not difference compare…”

27. Line 164, wrong usage of semi-colon.

Fixed.

28. Line 166, "we determined the lowest concentration of adenosine capable of increasing EBV lytic protein production", can the author use another term to replace "increasing", "increasing" here does not have a meaning.

Fixed. We revised the sentence “Thus, we determined the lowest concentration of adenosine to initiate the EBV lytic protein production.”

29. Line 187, "up to 250-500-fold..." the author can directly write "approximately 500 fold" as supported by Figure 4F.

Fixed.

30. Line 190, the author can directly write ADORA1 involved instead of "an adenosine receptor"

Fixed.

31. Line 234, "These results suggested that"

Fixed.

32. Line 245, the author should name the "previously identified adenosine receptors" directly here.

We revised “adenosine receptors”.

33. Line 249-250, the author should name the concentration of DPCPX used instead of saying "ten-fold serial dilutions of DPCPX".

Fixed. We marked DPCPX concentration as “0.05, 0.5, and 5 μM DPCPX” at line 207.

34. Line 257, "Thus, DPCPX could partially inhibit"

Fixed.

35. Line 277, "TRCN00008038 (ADORA1-8038 shR) downregulated ADORA1 most efficiently"

Fixed.

36. Difficult to understand, line 285 "RT-qPCR analysis determined what effect adenosine did cause on EBV gene transcription"

We revised the sentence “Next, we determined transcription level of EBV genes and ADORA1 using RT-qPCR analysis.”

37. Difficult to understand in lines 308-309, 331, 334-335.

Fixed.

38. Line 337, "decreased cAMP levels for 2.8 fold..."

Fixed.

39. Line 337, "This result indicatethat adenosine was more likely to initiate..."

Fixed.

40. Line 339, "a key role in EBV lytic reactivation through BZLF1 upregulation" remove "the"

Fixed.

41. Line 343, “The assay was designed as described previously..."

Fixed.

42. Line 345, “drinking water were administered orally..."

Fixed.

43. Line 347, “measured the weights..."

Fixed.

44. Line 347, "they" refers to?

We revised “they” to “the weights”.

45. Line 348, “were not changed during the whole experimental period"; "We then measured thesizes..."

Fixed.

46. Line 351, "Tumors were removed from mice and were subjected to..."

Fixed.

47. Difficult to understand, line 353-355, "However..."

Fixed.

48. Line 355, can the author replace "work for” to another word.

We revised word from ‘work for’ to ‘influence’.

Reviewer 2 Report

Here, Choi and others present that adenosine treatment leads to viral lytic state in an EBV-associated gastric carcinoma cell line. They paid significant efforts for this manuscript, but one major weakness is the authors used only one cell line for in vitro assays. For generalization, testing other cell lines is expected.

Overall, Figure legends must be elaborated so that readers can follow easily.

Page 2 line 89-90; Qp and Fp are specific promoters for EBNA1, but not “latent genes” nor “EBV lytic genes” in general. Must be corrected.

Fig. 1E, Fig. 4F Ext.; for experiments like this, the supernatant must be treated with DNase before purification and extraction of virus particles, in order to eliminate naked viral genome and unsuccessfully packaged viral DNA.

Page 10 line 254; “but few were suppressed by DPCPX”.

Page 10 line 257; “were not affected by DPCPX”.

Is this really what the authors meant?

Page 10 line 271; “transfected” must read “infected” or “transduced”.

I suggest the authors include supplemental Fig1 as a part of Fig. 9. Otherwise, decrease of cAMP level by adenosine cannot be understood intuitively.

Page 14 line 379-392; if difference in cell lines can account for the difference between their previous paper (ref13) and ref 12, the authors in this paper should test other cell lines, such as AGS-EBV.

Author Response

ere, Choi and others present that adenosine treatment leads to viral lytic state in an EBV-associated gastric carcinoma cell line. They paid significant efforts for this manuscript, but one major weakness is the authors used only one cell line for in vitro assays. For generalization, testing other cell lines is expected.

 Overall, Figure legends must be elaborated so that readers can follow easily.

 Thank for your advice. We added the each concentration and incubation times. Also, we marked statistical analysis to understand easily.

Page 2 line 89-90; Qp and Fp are specific promoters for EBNA1, but not “latent genes” nor “EBV lytic genes” in general. Must be corrected.

In EBV genome, four promoters, Cp, Wp, Qp, and Fp, are known to drive EBNA1 expression, respectively. In general, C promoter is related with EBV lytancy I and F promoter is related with EBV lytic replication. Also, EBNA1 transcripts are known to originate from four different promoters and depend on the EBV life cycles, latency or lytic. Cp and Wp also known the EBNA promoters and the alternative EBNA promoter Qp is also used in type of EBV latency.

REF: Tao Q, Robertson KD, Manns A, Hildesheim A, Ambinder RF. The Epstein-Barr virus major latent promoter Qp is constitutively active, hypomethylated, and methylation sensitive. J Virol. 1998;72(9):7075-83. 

REF: Masucci, M.G.; Contreras-Salazar, B.; Ragnar, E.; Falk, K.; Minarovits, J.; Ernberg, I.; Klein, G. 5-Azacytidine up regulates the expression of Epstein-Barr virus nuclear antigen 2 (EBNA-2) through EBNA-6 and latent membrane protein in the Burkitt’s lymphoma line rael. J. Virol. 1989, 63, 3135–3141.

Fig. 1E, Fig. 4F Ext.; for experiments like this, the supernatant must be treated with DNase before purification and extraction of virus particles, in order to eliminate naked viral genome and unsuccessfully packaged viral DNA.

 Thank for your mention. After ultracentrifuge step, we treated DNase I for 30 min at 37 in virus particles and then incubate for 10 min at 75 to inactivate the enzyme activity. We marked material and methods. 

Page 10 line 254; “but few were suppressed by DPCPX”.

Page 10 line 257; “were not affected by DPCPX”.

Is this really what the authors meant?

We checked and revised sentences at line 216-223.

Transcription levels of most EBV gene were increased by adenosine and those of EBV genes markedly were reduced by co-treatment with adenosine and DPCPX. Transcription levels of ADORA1 and 7 EBV genes were significantly enhanced by adenosine and LMP1, EBER1, and EBNA1 were fully reduced by co-treatment with DPCPX and adenosine (p=n.s, respectively). Although LMP2, BLLF1, BZLF1, and BRLF1 did not reduced completely by co-treatment with DPCPX and adenosine, co-treatment with DPCPX and adenosine has the ability to reduce EBV gene transcriptions which were increased by adenosine treatment.

Page 10 line 271; “transfected” must read “infected” or “transduced”.

 Fixed. We changed ‘infected’

I suggest the authors include supplemental Fig1 as a part of Fig. 9. Otherwise, decrease of cAMP level by adenosine cannot be understood intuitively.

Thank for your suggestion. We revised Fig.9. ATP and cAMP data was replaced in Fig.9 and ATP and cAMP standard curve were replaced into suppl.Fig.1. In case of cAMP assay, we determined which adenosine receptors are highly related with EBV lytic reactivation by adenosine treatment. Fig. 6A showed that adenosine induce both ADORA1 and ADORA2A in EBVaGC. We checked weather ADORA1 or ADORA2A mainly acts as EBV lytic reactivation by adenosine. While ADORA1 inhibits the switch over from ATP to cAMP by downregulation of adenylyl cyclase, ADORA2A activates adenylyl cyclase and induces intracellular cAMP ratio. Therefore, we confirmed intracellular cAMP ratio from EBVaGC treated with dose-dependent adenosine.        

Page 14 line 379-392; if difference in cell lines can account for the difference between their previous paper (ref13) and ref 12, the authors in this paper should test other cell lines, such as AGS-EBV.

Since we heard that AGS-EBV cells used in academic community was infected other RNA virus, we was careful in using AGS-EBV cells in this paper. So, we used AGS-EBV cells in minimum amount where are in Fig.4C and Fig6D.

Reviewer 3 Report

Sun Jin Choi at al provide a comprehensive account of Cordyceps miltaris (CME) bioactive compounds Adenosine and Cordycepin mediated lytic induction of EBV SNU-719 cells. They further explore if the Adenosine-ADOR1 axis is involved in this signaling mechanism where EBV is lytic induced in excess of Adenosine. They also perform some animal work to show that adenosine may reduce tumor size in the mice injected with EBV infected cells.

Minor

Line 41: Sentence and follow-up sentence need to get rewarded. It can be implied differently. In short, strict latent infections phase does not believed to produce virus.

Line 44 and Lines 49-54 need accurate reference. A lot of original work need citations here.

Use original research paper for citation 14 in line 70

Major:

Fig 1B: How were the band densities measured? What system was used? How is the number obtained? This should be reflected in the figure legend/results section. Also loading control bands should also be represented in number form.

How long this treatment experiment was performed?

Fig. 1C: Molecular weight marker is missing? Indicate size of the proteins is missing

1D. Explain the color coding of the text box used for gene display. Annotate Fp promoter and genes under its control.

Fig. 2A. Scales are invisible in fig. 2A. Mark the names of the compound with number?

Fig, 2B. Explain the variations in CD50 values for COR and ADE of current research vs previously published studies by Ryu, et al? How long the cells were treated for?

Fig. 3 A and 3B: Explain the differences observed in gene expression and protein production profiles for ADE and COR

Fig. 4 A, B and C: What are the reasons for band differences in for Zta? Again time used for treatment should be mentioned in legend.

Fig, 4 D and E: What scale the magnification is shown for IFA? Can it be shown at higher magnification? It is not clear in the figure if Zta expressed here is nuclear or cytoplasmic?

4. F: Why are the episome number down at ADE 820 uM?

Fig. 10:

How many animals used? What was the concentration of ADE used All this should be mentioned in fig legend. Authors should include the images of tumors that were measured.

What if the ADE is withdrawn from the treatment? ? Do the animals survive long term? What if the ADE is withdrawn from the treatment?

Are the authors claim that Zta is directly targeted by ADOR1 receptor mediated signaling? The molecular underpinning of Zta promoter may also have indirect link? May need to do luciferase assay to prove this?

Finally, Adenosine has been shown to induce apoptosis. Did authors looked into this angle of ADE mediated apoptosis: Ref: https://www.ncbi.nlm.nih.gov/pubmed/26494951 https://www.tandfonline.com/doi/abs/10.1080/14737140.2017.1316197?journalCode=iery20

Authors end their discussion with a sentence that can be mis- interpreted by general public. This reviewer advise to reformat this sentence..

Author Response

Sun Jin Choi at al provide a comprehensive account of Cordyceps miltaris (CME) bioactive compounds Adenosine and Cordycepin mediated lytic induction of EBV SNU-719 cells. They further explore if the Adenosine-ADOR1 axis is involved in this signaling mechanism where EBV is lytic induced in excess of Adenosine. They also perform some animal work to show that adenosine may reduce tumor size in the mice injected with EBV infected cells.

Minor

Line 41: Sentence and follow-up sentence need to get rewarded. It can be implied differently. In short, strict latent infections phase does not believed to produce virus.

Thank for your advice. We checked and revised sentences.

Line 44 and Lines 49-54 need accurate reference. A lot of original work need citations here.

Thank for your mention. We checked and replaced reference in line 41-45.

Use original research paper for citation 14 in line 70

We added two papers mentioned the role of adenosine deaminase in reducing the biological activities of cordycepin.

16.      Cristalli, G.; Costanzi, S.; Lambertucci, C.; Lupidi, G.; Vittori, S.; Volpini, R.; Camaioni, E., Adenosine deaminase: functional implications and different classes of inhibitors. Med Res Rev 2001, 21, (2), 105-28.

17.      Spiers, A. S.; Parekh, S. J.; Bishop, M. B., Hairy-cell leukemia: induction of complete remission with pentostatin (2'-deoxycoformycin). J Clin Oncol 1984, 2, (12), 1336-42.

18.      Li, G.; Nakagome, I.; Hirono, S.; Itoh, T.; Fujiwara, R., Inhibition of adenosine deaminase (ADA)-mediated metabolism of cordycepin by natural substances. Pharmacol Res Perspect 2015, 3, (2), e00121.

Major:

Fig 1B: How were the band densities measured? What system was used? How is the number obtained? This should be reflected in the figure legend/results section. Also loading control bands should also be represented in number form.

We used Image J program to measure the band intensity.

How long this treatment experiment was performed?

We are sorry for our mistake. Western blot, immune-fluorescence, RT-qPCR, promoter usage assay and ATP and cAMP assay were performed for 48 h treatment. Intracellular and extracellular EBV copy number assay were proceeded for 72 h. We marked incubation time and treatment concentration in all figure legends.

Fig. 1C: Molecular weight marker is missing? Indicate size of the proteins is missing

We are sorry for our mistake. We marked protein size in all figures.

1D. Explain the color coding of the text box used for gene display. Annotate Fp promoter and genes under its control.

In Fig.1D, we choose 2 types of color, orange and blue. Orange color mean latency genes and blue color mean lytic genes in EBV genome. In EBV lytic reactivation, gene expressions of latency and lytic phase increase. Because the Fp promoter is related for EBV lytic reactivation, EBV lytic genes marked blue color. 

 Fig. 2A. Scales are invisible in fig. 2A. Mark the names of the compound with number?

We checked data and marked compound name and X and Y axis.

Fig, 2B. Explain the variations in CD50 values for COR and ADE of current research vs previously published studies by Ryu, et al? How long the cells were treated for?

 We treated for 48 h to measure the CD50 values using CCK-8 kit.

Fig. 3 A and 3B: Explain the differences observed in gene expression and protein production profiles for ADE and COR

We added the differences between adenosine and cordycepin at line 136-137 and 140-141.

This results show that adenosine upregulates EBV gene transcriptional level except EBNA3C, but cordycepin induce only 2 EBV gene transcription including EBER1 and BZLF1.

Consistent with Fig. 3A, cordycepin also slightly enhanced EBV protein production, but this enhancement was weaker than that of adenosine.

Fig. 4 A, B and C: What are the reasons for band differences in for Zta? Again time used for treatment should be mentioned in legend.

We marked treatment time in figure legend.

Fig, 4 D and E: What scale the magnification is shown for IFA? Can it be shown at higher magnification? It is not clear in the figure if Zta expressed here is nuclear or cytoplasmic?

We had some difficulties in magnifying Zta foci detected on SNU719 cells.

One of the difficulties might be the morphology of SNU719 cells. SNU719 cells is round-shape and agglomerates to grow up in culture plate. So, it was difficult to separate the localization of nuclear and cytoplasm by immuno-fluorescence assay.

As seen in Supplemental Figure 2, we additionally conducted immunocytochemistry assay (ICA) to reconfirm that lower concentration of adenosine could induce Zta expression. Anti-Zta antibody was detected in SUN719 cells using DAB substrate kit which could react with HRP conjugated in Goat anti-mouse IgG antibody. Zta expression was getting stronger as we treated SNU719 cells with higher concentrations of adenosine. This result was likely to support the BZLF1 induction by lower concentrations of adenosine presented in Fig 4D.

4. F: Why are the episome number down at ADE 820 uM?

We are sorry for our mistake for handling experiments. To support our hypothesis and Fig.4 results, we measured intracellular and extracellular EBV copy number by adenosine-dose dependent manner. We replaced Fig. 4F graph. 

Fig. 10:

How many animals used? What was the concentration of ADE used All this should be mentioned in fig legend. Authors should include the images of tumors that were measured.

Thank for your mention. We used 6 for each xenograft mouse group. Adenosine concentration is 30 mg/kg/day. We marked mouse number and adenosine concentration in figure legend.

We added xenograft mouse model at Fig. 10D.

What if the ADE is withdrawn from the treatment? ? Do the animals survive long term? What if the ADE is withdrawn from the treatment?

It is good question. But we cannot measured the effect and survival for ADE withdrawn. Fig.10E shown that adenosine-treated xenograft tissues expressed Zta, EBV lytic protein. Fig. 10C shown that xenograft tumor size are decreased after 7 day of adenosine treatment. Synthetically, we cautiously guess that ADE withdrawn after sufficient EBV lytic reactivation might affect for anti-EBVaGC.   

Are the authors claim that Zta is directly targeted by ADOR1 receptor mediated signaling? The molecular underpinning of Zta promoter may also have indirect link? May need to do luciferase assay to prove this?

 We did not check interaction between Zta and ADORA1 by directly of indirectly. Several papers showed that EBV lytic phase lead to apoptosis and adenosine induce cancer cell death. So, we guessed that adenosine-dependent ADORA1 expression would enhance apoptosis in EBVaGC.

Finally, Adenosine has been shown to induce apoptosis. Did authors looked into this angle of ADE mediated apoptosis: Ref: https://www.ncbi.nlm.nih.gov/pubmed/26494951 https://www.tandfonline.com/doi/abs/10.1080/14737140.2017.1316197?journalCode=iery20

EBV lytic reactivation paralleled activation of apoptosis. Several papers shown that EBV lytic reactivation is closely related with apoptotic cell death. But, all types of stimulant of apoptosis cannot induce EBV lytic reactivation. Therefore, It is important to identify the compounds which have properties for anti-EBV or anti-EBV associated malignancies.

First we checked CD50 concentration from EBVaGC and those of CD50 were used for our experiments. Fig. 2B shown that the CD50 were 375 μM for cordycepin, 820 μM for adenosine, 105 μM for guanosine, and 500 μM for uridine, respectively. All high concentration of compounds induce cell death but only adenosine and cordycepin induce EBV lytic reactivation (Fig. 3A). So, we focus that which compounds in CME have properties for EBV lytic reactivation and anti-EBVaGC. Our study suggested that CME induce EBV lytic reactivation and this effect involved adenosine rather than cordycepin. Therefore, adenosine have potential as a novel compound for anti-EBV or anti-EBVaGC.

 Authors end their discussion with a sentence that can be mis- interpreted by general public. This reviewer advise to reformat this sentence.

We are very sorry for our mistake. We checked sentence and revised.